# Cell type-specific transcriptomics of hypothalamic energy-sensing neuron responses to weight-loss

**Fredrick E Henry[1†], Ken Sugino[1†], Adam Tozer[2], Tiago Branco[2], Scott M Sternson[1]***

[1]Janelia Research Campus, Howard Hughes Medical Institute, Ashburn, United States; [2]Division of Neurobiology, Medical Research Council Laboratory of Molecular Biology, Cambridge, United Kingdom

**Abstract** Molecular and cellular processes in neurons are critical for sensing and responding to energy deficit states, such as during weight-loss. Agouti related protein (AGRP)-expressing neurons are a key hypothalamic population that is activated during energy deficit and increases appetite and weight-gain. Cell type-specific transcriptomics can be used to identify pathways that counteract weight-loss, and here we report high-quality gene expression profiles of AGRP neurons from well-fed and food-deprived young adult mice. For comparison, we also analyzed Proopiomelanocortin (POMC)-expressing neurons, an intermingled population that suppresses appetite and body weight. We find that AGRP neurons are considerably more sensitive to energy deficit than POMC neurons. Furthermore, we identify cell type-specific pathways involving endoplasmic reticulum-stress, circadian signaling, ion channels, neuropeptides, and receptors. Combined with methods to validate and manipulate these pathways, this resource greatly expands molecular insight into neuronal regulation of body weight, and may be useful for devising therapeutic strategies for obesity and eating disorders.

**\*For correspondence:** sternsons@janelia.hhmi.org

[†]These authors contributed equally to this work

**Competing interests:** The authors declare that no competing interests exist.

## Introduction

Neurons that express *Agouti related protein* (*Agrp*) and *Proopiomelanocortin* (*Pomc*) comprise two intermingled molecularly defined populations in the hypothalamic arcuate nucleus (ARC) that mediate whole-body energy homeostasis in conjunction with other cell types. AGRP and POMC neurons positively and negatively regulate body weight, respectively (*Aponte et al., 2011*; *Krashes et al., 2011*). AGRP neurons transduce circulating signals of energy deficit into increased output of the neuropeptides AGRP and Neuropeptide Y (NPY) as well as the neurotransmitter γ-aminobutyric acid (GABA), each of which contribute to increased appetite and body weight. Correspondingly, weight loss is accompanied by increased *Agrp* and *Npy* gene co-expression in AGRP neurons (*Hahn et al., 1998*), as well as increased electrical activity (*Takahashi and Cone, 2005*) and synaptic plasticity (*Yang et al., 2011*; *Liu et al., 2012*). In contrast, during energy deficit, POMC neurons decrease electrical activity due to inhibitory synaptic input from AGRP neurons (*Takahashi and Cone, 2005*; *Atasoy et al., 2012*), and *Pomc* neuropeptide gene expression is reduced (*Schwartz et al., 1997*). AGRP and POMC neurons are thus both associated with sensing and counteracting energy deficit states. Because these neurons play major reciprocal roles in energy homeostasis, investigations of the molecular response pathways for AGRP and POMC neurons to weight-loss are critical for identifying key control points associated with regulation of body weight.

AGRP and POMC neurons sense energy deficit, in part, through responding to the metabolic hormones ghrelin, leptin, and insulin. Signaling pathways downstream of the receptors for these hormones have been elucidated (*Banks et al., 2000*; *Kitamura et al., 2006*), but most of the other molecular processes involved in the cellular response to systemic metabolic challenge in AGRP and POMC neurons remain unexplored. In light of this, a transcriptome-wide view of gene expression

**eLife digest** Humans and other animals must get adequate nutrition in order to survive. As a result, the body has several systems that work side by side to maintain a healthy body weight and ensure that enough food gets eaten to provide the energy that the body needs. Problems with these systems can contribute towards obesity and other eating disorders.

Certain types of cells in the brain play important roles in controlling weight and appetite, although the genes and cellular mechanisms that underlie these abilities are not well understood. When an animal is deprived of food, so-called AGRP neurons produce molecules that increase appetite and make it easier to gain weight. These neurons also go through structural changes and increase their electrical activity during weight loss. Another group of cells, called the POMC neurons, becomes less active when an animal is deprived of energy.

Using a technique called cell type-specific transcriptomics, Henry, Sugino et al. have now revealed that the expression of hundreds of genes in AGRP and POMC neurons changes depending on whether mice are well fed or food deprived. Food deprivation also affects more genes in AGRP neurons than has been seen in other types of brain cell, and the AGRP neurons are also more sensitive to a change in food intake than POMC neurons.

In the future, this gene expression data and knowledge of the pathways affected by the genes could help researchers to develop new treatments for obesity and other disorders that affect appetite. Henry, Sugino et al. then mapped how these changes in gene expression trigger molecular "pathways" in the neurons that alter how the cells work. These affect many parts of the cells, including ion channels, transcription factors, receptors, and secreted proteins. In addition, food deprivation activated pathways in AGRP neurons that protect the cells from damage and death caused by elevated neuron activity and also triggered signaling pathways that increase body weight.

In the future, this gene expression data and knowledge of the pathways affected by the genes could help researchers to develop new treatments for obesity and other disorders that affect appetite.

changes can provide a foundation for investigating the neuronal cell biology of these energy homeostasis sensing neurons during a state of energy deficit.

The transcriptional response to food-deprivation has been reported previously using tissue samples from the entire hypothalamus (*Guarnieri et al., 2012*) or ARC (*Li et al., 2005*; *Jovanovic et al., 2010*), but these studies lacked cell type-specificity necessary to understand the molecular response properties of individual neural circuit nodes. Recent approaches employing immunoprecipitation of messenger RNA (mRNA) in molecularly defined and even projection-specific populations (*Heiman et al., 2008*; *Dalal et al., 2013*; *Ekstrand et al., 2014*; *Allison et al., 2015*) require large numbers of cells, and therefore have been challenging to perform for neurons with small population sizes, such as AGRP and POMC neurons. A transcriptional profile of AGRP neurons has been obtained previously from dissociated tissue in which fluorescently labeled AGRP neurons were sorted and pooled from ~40 neonatal mice and compared to a similar number of neonatal AGRP neuron-specific *Foxo1* knockout mice (*Ren et al., 2012*). In neonatal mice, cells are readily dissociated, but AGRP neurons are not necessary for early neonatal life and their axons are not developed (*Bouret et al., 2004*; *Luquet et al., 2005*), thus, the relevance of neonatal gene expression patterns to those in adult mice is uncertain. Moreover, comparing only one sample from two conditions prevents statistical analysis of differentially expressed genes (DEG). Recent technical improvements in cell sorting and transcriptional profiling methods have enabled the generation of high quality gene expression profiles from small numbers of fluorescently labeled neurons (typically 40–250 neurons) from single adult mouse brains (*Sugino et al., 2006*; *Okaty et al., 2011*). Importantly, this permits use of individual animals as replicates for comparing gene expression profiles under different conditions, which is the approach that we used here.

We performed RNA sequencing (RNA-Seq) using AGRP and POMC neurons from ad libitum fed young adult mice as well as from mice after 24-hr food deprivation. We confirmed a small number of previously reported changes in gene expression, and also identified hundreds of additional DEG. These changes in gene expression allowed identification of coordinated signaling pathways that are

involved in the response to food deprivation, and we focus here on neuropeptides, G-protein coupled receptors (GPCRs), as well as pathways associated with neuron electrical activity, circadian regulation, and endoplasmic reticulum (ER)-stress signaling. This resource, which includes gene expression profiles for AGRP and POMC neurons as well as methods for validating and evaluating the functional significance of these changes, provides a foundation for in depth analysis of the molecular control points for neuron populations involved in energy homeostasis.

## Results

We obtained transcriptional profiles from AGRP and POMC neurons from male young adult mice (6.5–8 weeks old) under ad libitum fed (fed, AGRP, POMC: n = 5 mice each) or 24-hr food deprived (FD, AGRP: n = 6 mice, POMC: n = 5 mice) conditions during the middle of the light phase (percent weight change AGRP.fed: 0.4 ± 1.5%, POMC.fed: 1.0 ± 0.9%, AGRP.FD: −19.4 ± 1.8%, POMC.FD: −17.7 ± 1.6%; mean ± sd). AGRP and POMC neurons were dissociated and manually sorted from the hypothalamic ARC of $Npy^{hrGFP}$ and $Pomc^{topazFP}$ transgenic mice, respectively, where the neurons could be identified by fluorescent protein expression (*Figure 1A*). Nearly all (94.9%) ARC$^{NPY}$ neurons are reported to express *Agrp*, which is selectively expressed in this brain area (*Broberger et al., 1998*), therefore we refer to these cells as AGRP neurons. Each pool of sorted fluorescent neurons (mean ± sd: 102 ± 40.5 neurons, range: 44–214 neurons) from a single brain was used for a separate sample from which RNA was extracted and then reverse transcribed, amplified, sequenced, and analyzed (*Figure 1A*, *Figure 1—figure supplement 1A–C*, and *Supplementary file 1*).

Manual sorting of fluorescently labeled cells has been previously compared with other cell type-specific RNA isolation methods and shown to yield similar or improved RNA-Seq data quality (*Okaty et al., 2011*). In particular, manual-sorting can provide high purity samples (*Okaty et al., 2011*), which we initially assessed by comparing neuropeptide markers between AGRP and POMC neuron populations. The transcript *Npy* was enriched by 432-fold in AGRP neurons over POMC neurons, and *Agrp* was 252-fold enriched; conversely, *Pomc* was enriched by 351-fold in POMC neurons (*Figure 1—figure supplement 1D*). We also checked for non-neuronal contamination by looking at highly expressed marker genes for astrocytes, myelin oligodendrocytes, microglia, and endothelial cells. A prior effort to obtain transcriptomic profiles from manually sorted AGRP neurons (*Ren et al., 2012*) showed evidence of differential oligodendrocyte cell contamination in the two samples used for analysis (*Figure 1—figure supplement 1E*). The differential expression for these primarily non-neuronal markers in our AGRP and POMC neuron samples relative to purified samples of these non-neuronal cell types was at least 300-fold (range: 318- to 8904-fold), indicating a high level of sample purity (*Figure 1B*, *Figure 1—figure supplement 1F*).

AGRP and POMC neurons are functionally distinct cell types, and gene expression profiles can be used to measure their molecular differences (*Figure 1C*). Multidimensional scaling (MDS), which is a method for visualization of similarity in gene expression patterns, showed clear separation between AGRP and POMC neuron populations regardless of deprivation state, and AGRP neurons, but not POMC neurons, were also distinct after food deprivation (*Figure 1D*). The AGRP and POMC neuron cell types from fed mice showed substantial differences, with 694 DEG (*Figure 1E* and see 'Materials and methods' for criteria). Consistent with this difference between AGRP and POMC neurons, clear separation of the cell types by MDS was maintained after exclusion of *Agrp*, *Npy*, and *Pomc* (*Figure 1D*), the top 30-most DEG, or even all DEG (*Figure 1—figure supplement 1G*). Therefore, transcriptional profiles indicate that these canonical markers are not required to distinguish cellular identity of AGRP and POMC neurons.

Both AGRP and POMC neurons have been reported to show alterations of gene expression with food deprivation, but the global transcriptomic response of these two cell populations to an energy deficit state is not known. As expected, *Agrp* (+4.4-fold, q = 3.3e$^{−6}$ [q-value is false discovery rate-corrected p-value]) and *Npy* (+3.3-fold, q = 1.5e$^{−7}$) were upregulated and *Pomc* expression modestly decreased (−1.6-fold, q = 0.038) (*Figure 1—figure supplement 1D*). In addition, members of the *Fos* immediate-early-gene family were upregulated selectively in AGRP neurons from FD mice (*Figure 1—figure supplement 2A*), consistent with elevated AGRP neuron activity in energy deficit (*Takahashi and Cone, 2005*), and the selectivity of this response also confirmed that dissociation and sorting did not result in cellular activation, similar to a previous study (*Okaty et al., 2011*). Using our transcriptomic data, we found that gene expression changes in AGRP neurons were much more extensive than for POMC neurons in response to food-deprivation. AGRP neurons showed 826 DEG after food deprivation (51.8% upregulated), but only 47 genes (46.8% upregulated) for POMC

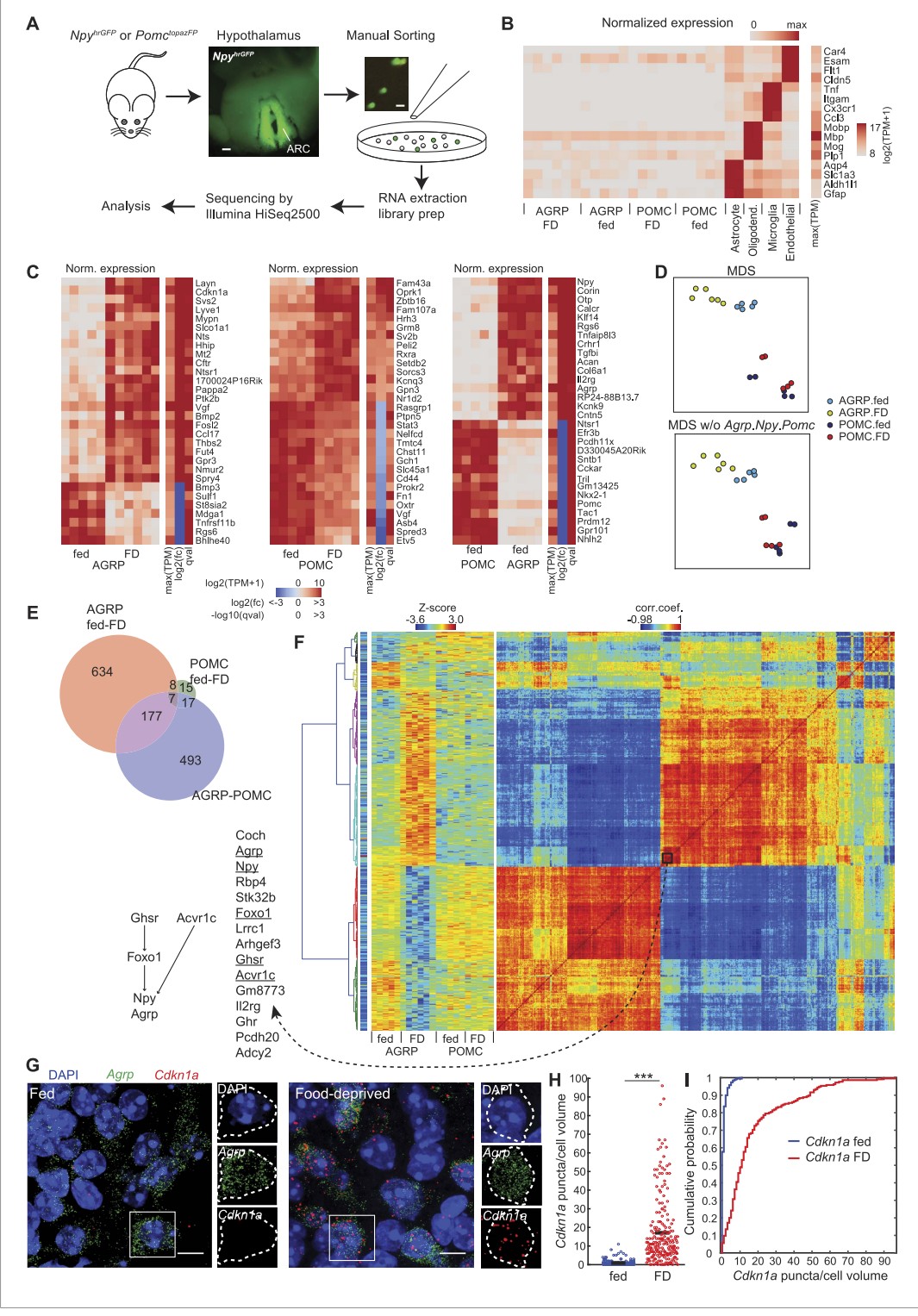

**Figure 1**. Cell type-specific transcriptomic profiling of starvation-sensitive neurons. (**A**) Schema for dissection and sorting of fluorescent neurons from the hypothalamic arcuate nucleus (ARC) followed by cell type-specific RNA-Seq. Scale: ~200 μm (left), ~20 μm (right). (**B**) Expression levels of marker genes for astrocytes, myelin oligodendrocytes, microglia, and endothelial cells indicate high purity of AGRP and POMC samples. FD: 24-hr food-deprived. Left, for each sample, log expression levels for a single gene in each row normalized by maximum expression level of the transcript in any of the samples. Right, sidebar shows maximum expression level for each row (each transcript) across

*Figure 1. continued on next page*

*Figure 1. Continued*

all samples. TPM: transcripts per million. (**C**) Top 30 differentially expressed genes (DEG) for: AGRP neurons, FD/fed; POMC neurons, FD/fed; AGRP/POMC neurons (fed). FD: 24-hr FD. As in (**B**), each row corresponds to a transcript where the expression level in each sample is normalized by maximum expressions on level in the row. The sidebars show maximum expression level [max(TPM)] across samples in each row (each transcript). In addition, log2(fold-change) [log2(fc)], and q-value (qval) for the differential expression across fasted/fed states (left and middle) or AGRP/POMC expression levels are shown. (**D**) Top, multidimensional scaling (MDS) projection of distance (1-corr.coef.) between samples. Bottom, MDS without *Agrp*, *Npy* and *Pomc* genes in the calculation. (**E**) Venn diagram for DEG between FD and fed conditions (AGRP FD-fed: red, POMC FD-fed: green) and between AGRP and POMC neurons (both fed, purple). Reported DEG required q-value <0.05, abs[log2(fc)] > 1 and mean CPM > 20 in at least 1 cell type/condition (see 'Materials and methods'). (**F**) Hierarchical clustering of DEG. Matrix in the middle indicates standardized expression level for the samples (columns) and DEG (rows). The matrix on the right shows the correlation coefficients between genes, calculated across all samples. The colormap on the left indicates maximum TPM expression level in log2 scale, which ranges from 2.96 (blue) to 15.6 (red). Left, genes in a cluster that comprise known pathways that regulate *Agrp* and *Npy* expression. (**G**) Representative images of double single molecule fluorescence in situ hybridization (smFISH) for *Agrp* and *Cdkn1a*. Scale, 10 μm. (**H, I**) Population counts (bars: mean value) (**H**) and cumulative probability distributions (**I**) of *Cdkn1a* puncta per cell volume in AGRP neurons (p = $1.5e^{-53}$, Kolmogorov–Smirnov [ks]-test). ***p < 0.001. Fed, n = 189 cells; FD, n = 215 cells; 3 mice per condition.
The following figure supplements are available for figure 1:

**Figure supplement 1**. Comparison of AGRP and POMC neuron transcriptomic samples.

**Figure supplement 2**. DEG in AGRP and POMC neurons with food deprivation.

neurons, of which 15 genes were in common between the two cell types (*Figure 1C,E* and *Figure 1—figure supplement 2B*). This surprising 17-fold difference in the number of DEG between the two cell types indicates that AGRP neurons are regulated by deprivation states much more strongly than POMC neurons.

Hierarchical clustering of correlated gene expression patterns can reveal groups of genes that show similar responses to food-deprivation across different conditions and cell types. We identified many clusters of highly correlated genes based on their expression levels in AGRP and POMC neurons from fed and FD mice (*Figure 1F*). One small cluster of 15 genes included *Foxo1*, *Agrp*, and *Npy*, which are reported to be in the same pathway (*Kitamura et al., 2006*); moreover ghrelin receptor (*Ghsr*) and *Acvr1c*, both of which influence *Npy* expression (*Sandoval-Guzman et al., 2012*), were also highly correlated (*Figure 1F*). These data indicate that this resource can be a starting point for investigating patterns of gene regulation in AGRP and POMC neurons.

Our RNA-Seq data were also consistent with some results that have been obtained previously by transcriptional profiling of the hypothalamus without regard to cell type. For example, transcriptional profiling of whole-hypothalamus tissue samples indicated upregulation of the glucocorticoid-regulated gene *Cdkn1a* (*p21*) with food deprivation (*Guarnieri et al., 2012*; *Tinkum et al., 2013*). Here, we find that *Cdkn1a* is the second-most highly upregulated gene during food deprivation in AGRP neurons (+166-fold, q = $7.7e^{-7}$), but it is not significantly changed in POMC neurons (−1.5-fold, q = 0.79), illustrating the importance of cell type-specific neuronal profiling. We confirmed increased *Cdkn1a* expression in AGRP neurons from FD mice by double-label RNA single molecule fluorescence in situ hybridization (smFISH) (p = $1.5e^{-53}$, Kolmogorov–Smirnov (ks) test, *Figure 1G–I*). Similarly, microarray profiling of the ARC has shown *Asb4* was slightly downregulated (−1.25-fold) with food-deprivation (*Li et al., 2005*), but, with cell type-specific RNA-Seq we find that *Asb4* is strongly down-regulated selectively in POMC neurons from FD mice (POMC: −5.8-fold, q = $2.2e^{-9}$; AGRP: +1.1-fold, q = 0.71, *Figure 1C*). Finally, *Gpr17*, which has been reported to be expressed in AGRP neurons based on transcriptional profiling of cells isolated using fluorescence activated cell sorting (FACS) from neonatal mice (*Ren et al., 2012*) was not detected in our samples from either AGRP or POMC neurons from adult mice, possibly indicating developmental differences in gene expression or differences arising from oligodendrocyte cell contamination of neonatal samples. These comparisons illustrate the advantages of using high purity sorted cells from individual adult mice to allow for statistical analysis of gene expression from intermingled cell types under different conditions.

## Pathway analysis

In addition to changes in expression of individual genes and gene families such as kinases, phosphatases, and transcription factors that are evident from RNA-Seq data (*Figure 1—figure supplement 2C–E*), the coordinate regulation of multiple genes allows predictions to be made about pathways involved in the transition from energy replete to energy deficient physiological states. Gene annotation enrichment analysis highlighted several pathways that were significantly affected by food-deprivation (*Table 1*). One such previously established pathway in AGRP neurons is leptin receptor signaling. Leptin levels fall in energy deficit and this is associated with a concomitant rise in *Lepr* (+5.3-fold, q = $1.3e^{-10}$) in AGRP neurons, as was previously reported for whole-hypothalamus tissue samples (*Baskin et al., 1999*), but *Lepr* was not significantly changed in POMC neurons (−1.6-fold, q = 0.39). Consistent with low circulating leptin, which leads to reduced Lepr signaling, *Jak2*, the Lepr-associated kinase, and *Socs3*, a downstream target of Lepr signaling, were reduced in AGRP neurons (*Figure 1—figure supplement 2F,G*). Conversely, *Foxo1*, a transcription factor that is negatively regulated by leptin receptor signaling, was selectively increased in AGRP neurons with food deprivation.

Opposite to leptin, levels of the hormone ghrelin are elevated with food-deprivation. We found that *Ghsr* was upregulated in AGRP neurons after food-deprivation (+3.4-fold, q = $4.9e^{-9}$), but was nearly absent in POMC neurons (*Figure 1—figure supplement 2F,G*). Signaling components downstream of *Ghsr* were also upregulated, such as *Prkca*, a protein kinase C isoform. Therefore, our cell type-specific transcriptomic data recapitulates known leptin receptor and ghrelin receptor signaling pathways in AGRP and POMC neurons.

More importantly, though, this comprehensive transcriptomic resource can be used to identify pathways that have not been previously implicated in the physiological response to energy deficit in these cell types. We focused on investigating pathways that had not been examined in AGRP neurons, such as systems for ER-stress, circadian regulation, and synaptic function, as well as genes encoding ion channels, GPCRs, and secreted proteins.

## ER stress pathways

Gene annotation enrichment analysis (*Table 1*) highlighted, selectively in AGRP neurons, a transcriptional response for genes associated with ER-stress: the unfolded protein response (UPR) and ER-associated degradation (ERAD) of misfolded proteins. This pathway has not been examined previously in AGRP neurons; instead, based on whole hypothalamus analysis, UPR has been primarily associated with overnutrition states and leptin resistance (*Ozcan et al., 2009*), as opposed to the energy-deficit condition examined here. ER-stress responses occur during high levels of protein translation where unfolded proteins elicit a program of downstream transcriptional responses to increase protein folding and processing in the ER, and this pathway showed the most prominent effect on gene expression that we identified in AGRP neurons. This pattern of differential gene expression

**Table 1**. Gene annotation enrichment analysis of differentially expressed genes

| Pathway | −log(p-value) |
| --- | --- |
| Agouti related protein | |
| Leptin signaling | 4.9 |
| Glutamate signaling/Axonal guidance/Ephrin/Rho GTPase | 3.5 |
| Endoplasmic reticulum stress/Oxidative stress | 3.2 |
| G-protein coupled receptor signaling | 3.0 |
| Circadian rhythm signaling | 2.7 |
| Sperm motility | 1.8 |
| Proopiomelanocortin | |
| Gα$_i$ signaling | 6.3 |
| Tetrahydrobiopterin biosynthesis (*Gch1*) | 2.1 |
| Zymostrerol biosynthesis (*Msmo1*) | 1.8 |

was not observed in POMC neurons, indicating that it is not an artifact of the neuronal isolation procedure. Consequently we examined multiple aspects of ER-stress signaling by analysis of our RNA-Seq data, coupled with preliminary evaluation of ER-stress signaling using immunohistochemistry, a cell type-specific UPR pathway reporter, and smFISH.

ER-localized proteins were significantly overrepresented in the group of food-deprivation-regulated genes in AGRP neurons ($p = 8.3e^{-16}$, hypergeometric-test), and 95% of these DEG were upregulated (*Figure 2A*). A key UPR marker, *Hspa5*, which encodes the canonical ER-localized unfolded protein-sensing chaperone BiP, was selectively upregulated in AGRP neurons during food deprivation (AGRP: + 3.3-fold, $q = 2.5e^{-10}$; POMC: −1.2-fold, $q = 0.52$). Correspondingly, BiP-immunoreactivity was significantly increased by food-deprivation in AGRP neurons ($p = 2.5e^{-42}$, ks-test, *Figure 2B,D*), indicating UPR activation in AGRP neurons. In contrast, we also found that BiP-immunoreactivity was significantly reduced in POMC neurons during food-deprivation ($p = 2.5e^{-19}$, ks-test, *Figure 2C,D*), suggestive of reduced protein translational-load in this population.

UPR involves a program of gene expression that is regulated by *Ern1/Ire1*, which was selectively increased in AGRP neurons by food deprivation (+2.7-fold, $q = 0.001$). In response to the accumulation of unfolded proteins in the ER lumen, Ern1/Ire1 undergoes a conformational change, which activates endoribonuclease activity for the short-lived transcript *Xbp1* and splices it into a more stable mRNA, *Xbp1s*. These RNA-Seq data reveal increased abundance for the *Xbp1s* spliced mRNA in AGRP neurons after food-deprivation (*Xbp1s/Xbp1*, fed: 7.4 ± 1.0%, FD: 15.6 ± 3.8%, $p = 0.043$, one-tailed t-test; *Figure 2E*), consistent with activation of Ern1/Ire1 endonuclease activity in AGRP neurons after 24-hr food deprivation.

Xbp1s is a transcription factor that regulates UPR-related gene expression. Two well-established Xbp1s-dependent gene targets (*Lee et al., 2003*) were selectively upregulated in AGRP neurons after food deprivation: *Dnajc3/p58$^{IPK}$* (+1.9-fold, $q = 0.0007$) and *Dnajb9* (+2.2-fold, $q = 0.0003$). Many other genes regulated by Xbp1s were also increased in AGRP neurons during food-deprivation, including protein folding chaperones of the Hsp40 (Dnaj), Hsp70 (Hspa), and Hsp90 families, *Calr* and *Canx*, as well as protein disulfide isomerases (*Pdia3-6*), which aid in protein folding by catalyzing the formation of disulfide bonds (*Figure 2A*). *Atm* (−3.1-fold, $q = 7e^{-5}$), a DNA damage sensing-enzyme that, when reduced leads to increased *Xbp1* splicing (*He et al., 2009*), was downregulated with food deprivation. Taken together, Xbp1-splicing and the patterns of downstream gene expression indicate a previously unreported role for Ern1/Ire1 → Xbp1s signaling in the adaptive response of AGRP neurons to energy deficit.

We also investigated the involvement of two other UPR signaling pathways regulated by ER-bound transmembrane proteins, Eif2ak3 (also called PERK) and Atf6. The Eif2ak3/PERK (AGRP: −2.1-fold, $q = 0.23$) arm of the UPR pathway is the first to be engaged during ER stress and suppresses translation of most mRNA. We expected that prolonged translational arrest in AGRP neurons was unlikely because it is inconsistent with elevated neuropeptide production in AGRP neurons during energy deficit. Activation of Eif2ak3/PERK leads to mRNAs sequestration into messenger ribonucleoprotein particles that aggregate into stress granules containing the RNA binding protein, TDP43 (*Colombrita et al., 2009*). To assess stress granule formation, we used anti-TDP43 immunostaining. Elevated stress granule formation was not detected in AGRP neurons from FD mice ($p = 0.76$, ks-test, *Figure 2F,G*), which provides preliminary evidence that Eif2ak3/PERK-mediated translational arrest may not be engaged at this 24-hr food deprivation time-point. Moreover, transcripts for ER protein translocation (*Srp68, Srp72, Sec61a1, Sec61b1, Sec63, Serp1/Ramp4*) and Golgi trafficking (*Sec14l1, Sec22b, Sec24d*) were upregulated in AGRP neurons, possibly indicating increased protein translation and folding capacity during energy deficit and consistent with the requirement for increased peptidergic neurotransmission for AGRP neuron function.

An additional signaling arm of the UPR is mediated through the ER-bound protein Atf6 (AGRP: +2.0-fold, $q = 0.0016$). In response to ER stress, Atf6 is subject to proteolytic cleavage and releases an N-terminal transcription factor domain that translocates to the nucleus and initiates gene expression to increase protein folding. To examine the extent of Atf6 nuclear translocation in the response of AGRP neurons to food-deprivation, we developed a Cre recombinase-dependent N-terminal green fluorescent protein (GFP):Atf6 fusion (*Samali et al., 2010*) reporter of Atf6 cleavage for use in the brain (*Figure 2H*). After expression in AGRP neurons, we measured the nucleus:cytoplasm ratio of GFP, which was very low and was not significantly increased in FD vs ad libitum fed mice; whereas the ER-stress-inducer, tunicamycin, strongly increased nuclear localization of GFP in AGRP neurons (fed vs FD: $p = 0.44$, dimethyl sulfoxide (DMSO) vs

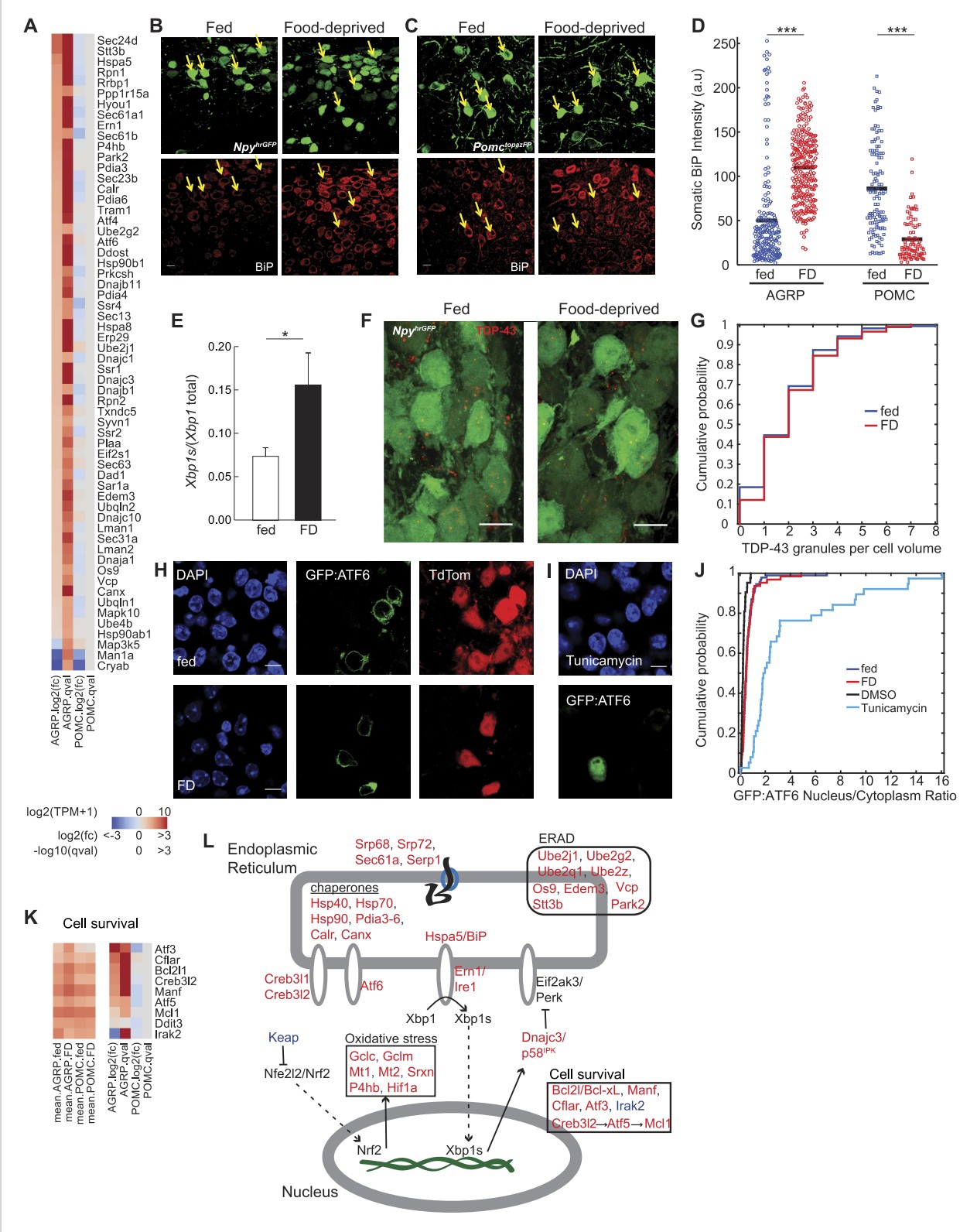

Figure 2. Food deprivation induces unfolded protein response in AGRP neurons. (A) Log2(fold-change) [log2(fc)] and q-values for genes associated with endoplasmic reticulum (ER) localization (KEGG pathway: mmu04141) that are affected by food deprivation in AGRP (left columns) or POMC (right columns) neurons. (B, C) Representative images showing BiP-immunofluorescence from $Npy^{hrGFP}$ or $Pomc^{topazFP}$ mice. Arrows: examples of fluorescently labeled (B) AGRP and (C) POMC neurons used for BiP quantification. Scale: 10 µm. (D) Population counts of BiP somatic intensity in AGRP or POMC neurons. AGRP.
*Figure 2. continued on next page*

*Figure 2. Continued*

fed, n = 209; AGRP.FD, n = 283; POMC.fed, n = 121; POMC.FD, n = 92; 3 mice per condition. Bars: mean values. Rank-sum test. ***p < 0.001. (**E**) Fraction of spliced to total *Xbp1* transcript isoforms in AGRP neurons. Unpaired one-tailed t-test. *p < 0.05. (**F, G**) Representative images (**F**) and cumulative probability distribution (**G**) of TDP43-immunoreactive-granules (red) in GFP-expressing AGRP neurons (p = 0.76, ks-test). Fed, n = 276; FD, n = 174; 2 mice per condition. Scale, 10 μm. (**H–J**) Representative images (**H, I**) of GFP:ATF6 expression *Agrp^Cre*;*ai9(tdtomato)* mice. Cumulative probability distribution (**J**) of the ratio of nuclear to cytoplasmic GFP fluorescence in AGRP neurons: fed vs FD (p = 0.44, ks-test) Fed, n = 92; FD, n = 63; 4 mice per condition; or AGRP neurons: DMSO vs tunicamycin (p = 2.9e⁻¹⁶, ks-test) DMSO, n = 42; Tunicamycin, n = 38; 1 mouse per condition. (**K**) Differentially expressed cell survival genes. Left, mean expression level [log2(TPM)] of each transcript of each experimental group. Right, log2(fold-change) and q-values for differential expression between FD and fed states separately for AGRP and POMC neurons. (**L**) Schematic for DEG in ER stress-associated pathways in AGRP neurons after food-deprivation. Red: upregulated expression, Blue: downregulated expression.

---

tunicamycin: $p = 2.9e^{-16}$, ks-test, *Figure 2H–J*). Low nuclear localization indicates minimal activation of the Atf6-signaling arm of UPR in AGRP neurons after 24-hr food deprivation. Consistent with this, a transcript selectively regulated by Atf6, *Herpud1* (*Lee et al., 2003*), was not significantly (q = 0.22) elevated in AGRP neurons from FD mice. However, because many Atf6-regulated genes are also regulated by Xbp1s, a robust ER-stress response can be maintained without Atf6 signaling (*Yamamoto et al., 2007*).

Oxidative stress pathways associated with elevated protein production are also increased in AGRP neurons during food deprivation. The transcription factor *Nfe2l2* (also called *Nrf2*) regulates oxidative stress and is normally rapidly degraded in the cytoplasm, in part through association with *Keap1* (*Kobayashi et al., 2004*), which shows reduced expression with food-deprivation (−2.2-fold, q = 0.05). Nfe2l2/Nrf2 upregulates expression of the key glutathione biosynthetic enzymes *Gclc* (+2.3-fold, q = 0.0002) and *Gclm* (+1.8-fold, q = 0.002), which are elevated in AGRP neurons with food-deprivation. Nfe2l2/Nrf2 increases other transcripts associated with oxidative stress, such as *Mt1* (+34-fold, q = 0.004), *Mt2* (79-fold, q = 0.004), and *Srxn1* (+17-fold, $q = 1.2e^{-6}$), which are among the most strongly upregulated transcripts in AGRP neurons during food-deprivation. Other oxidative stress transcripts were also upregulated, such as peroxiredoxin 2, *Prdx2* (+1.9-fold, p = 0.0004), the transcription factor *Hif1a* (+1.8-fold, p = 0.005) and the oxidative regulatory enzyme prolyl hydroxylase (*P4hb*, +2.7-fold, $p = 8e^{-7}$). Together, this group of cell type-selectively upregulated genes is consistent with an adaptive response to increased oxidative stress in AGRP neurons during energy deficit.

UPR also typically leads to ERAD of unfolded proteins. E2 ubiquitin conjugating enzyme subunits (*Ube2j1*, *Ube2g2*, *Ube2q1*, *Ube2z*, *Ube4b*) and the E3 ligase *Park2* were upregulated in AGRP neurons during food-deprivation (*Figure 2A*). *Os9*, a lectin that senses misfolded proteins, and *Edem3* and *Stt3b* (*Figure 2A*), which are enzymes that mark unfolded proteins for ERAD (*Sato et al., 2012*), were also increased. In addition, *Vcp*, a component for retrotranslocation of polyubiquitinylated unfolded proteins for degradation, was elevated in AGRP neurons during energy deficit. Collectively, this pattern of gene expression indicates engagement of ERAD during food-deprivation.

UPR is protective to cells for short periods of elevated ER-stress, but prolonged activation can result in apoptosis. The pro-apoptotic *Ddit3* (*Chop*) is typically upregulated during UPR, however *Ddit3/Chop* was unchanged in AGRP neurons after 24-hr food deprivation (*Figure 2K*). Instead, a variety of anti-apoptotic transcripts *Bcl2l* (*Bcl-xl*), *Manf*, *Mcl1*, and the caspase inhibitor *Cflar* were upregulated (*Figure 2K*). *Mcl1* upregulation is consistent with a previously established pathway Creb3l2 → Atf5 → Mcl1 (*Izumi et al., 2012*), each member of which is upregulated in food deprivation (*Figure 2K*). *Irak2* ($-4.7$-fold, $q = 3.7e^{-5}$) was strongly suppressed, consistent with reports that its reduction is protective against apoptosis (*Benosman et al., 2013*). In addition, the cellular stress-induced transcript *Atf3* (+15.5-fold, q = 0.005) also promotes neuron survival (*Francis et al., 2004*) and is increased with food deprivation. This pattern of gene expression provides preliminary evidence of a potential role for anti-apoptotic pathways in AGRP neurons during activation in response to energy deficit.

Taken together, cell type-specific transcriptional profiling, immunohistochemistry, and a cell type-specific UPR reporter construct indicate engagement of ER-stress pathways selectively in AGRP neurons during energy deficit. Increased neuron activity and elevated neuropeptide production associated with food-deprivation is expected to increase the translational-load in AGRP but not POMC neurons, and UPR may serve to cope with elevated neuropeptide and synaptic output. This is due, in part, to Xbp1s signaling and also results in induction of gene expression associated with

oxidative stress responses, ERAD, and protection of cells from apoptotic pathways that might otherwise be associated with prolonged ER-stress (*Figure 2L*).

## Regulation of circadian genes by food-deprivation

Expression of genes associated with circadian regulation was strongly altered by food-deprivation. We examined the expression levels of 19 core circadian reference genes (*Yan et al., 2008*; *Rey et al., 2011*) and found that after food-deprivation 9/19 of these genes were downregulated in AGRP neurons (*Bhlhe40* [*Dec1*], *Bhlhe41* [*Dec2*], *Nr1d1* [*Rev-erbα*], *Nr1d2* [*Rev-erbβ*], *Dbp*, *Hlf*, *Tef*, *Per2*, and *Per3*), and only 3/19 were upregulated (*Rorb*, *Nfil3*, *Per1*) (*Figure 3A*). For POMC neurons, only one transcript was significantly changed (*Nr1d2/Rev-erbβ*) (*Figure 3A*). The nine circadian genes downregulated in AGRP neurons during food-deprivation are established targets of two important circadian transcription factors, Clock and Arntl (also called, Bmal), which heterodimerize to regulate a large number of downstream genes through E-box transcriptional response elements (*Rey et al., 2011*). We analyzed a collection of E-box regulated genes that have been previously determined by Arntl (Bmal) chromatin binding, bioinformatic analysis, and transcriptional profiling (*Rey et al., 2011*). E-box-containing genes were over-represented among differentially expressed transcripts in AGRP neurons but not in POMC neurons (p = 0.00075 and 0.31, respectively; hypergeometric-test). For the E-box-containing genes that were significantly differentially expressed more than twofold in AGRP neurons after food-deprivation, 90% (18/20) were downregulated (*Figure 3B*). Conversely, in POMC neurons, *Nr1d2/Rev-erbβ* was the only E-box-containing gene that was significantly differentially expressed more than twofold, and it showed increased expression (*Figure 3B*). This indicates that differentially expressed E-box-containing genes during energy deficit are selectively reduced in AGRP neurons.

Many genes regulated by E-box transcriptional response elements show reduced expression in $Arntl^{-/-}$ mice (*Rey et al., 2011*), but *Clock* and *Arntl* expression levels in AGRP neurons were not significantly altered by food-deprivation. Immunohistochemistry for Arntl protein expression in AGRP neurons from fed and FD mice also showed similar levels (fed: 115 ± 4 a.u., n = 158 neurons; FD: 127 ± 4 a.u., n = 195 neurons; p = 0.058, rank sum test, *Figure 3—figure supplement 1*). An alternative pathway that has been shown to regulate E-box genes is the transcriptional splicing factor *Sfpq* (*Psf*) (*Duong et al., 2011*), which shows selectively increased expression in AGRP neurons after food-deprivation (AGRP: +2.0-fold, q = 0.00064; POMC: 1.0-fold, q = 1.0). However, detailed examination of pathways that regulate circadian E-box containing genes in AGRP neurons during energy deficit states is required.

## Synaptic function and plasticity

Excitatory synaptic plasticity occurs in AGRP neurons during energy deficit (*Yang et al., 2011*; *Liu et al., 2012*). Gene annotation enrichment analysis revealed significant changes in expression of genes associated with glutamate signaling, synaptic plasticity, and presynaptic function (*Table 1*). Moreover, genes encoding proteins that are localized to the synapse were over-represented in the DEG (q < 0.05) from AGRP neurons (p = $1.2e^{-6}$, hypergeometric-test; *Figure 3C*). For example, α-amino-3-hydroxy-5-methyl-4-isoxazolepropionic acid (AMPA) and kainate glutamate receptors, which mediate excitatory synaptic transmission, were upregulated in AGRP neurons (*Gria3*: +3.4-fold, q = $2e^{-5}$, *Grik1*: +3.0-fold, q = 0.03; *Grik3*: +3.2-fold, q = 0.005), but not in POMC neurons (see Ion Channels, below). Food-deprivation also induced upregulation of excitatory synaptogenic genes in AGRP neurons (*Syndig1*: +8.3-fold, q = 0.008; *Syndig1l*: +13.2-fold, q = 0.0001) (*Kalashnikova et al., 2010*; *Lovero et al., 2013*) as well as kinases (*Figure 1—figure supplement 2C*) that regulate activity-dependent changes in spine morphology and excitatory synaptic plasticity, such as *p21-associated kinase 3* (*Pak3*, +2.0-fold, q = $4e^{-7}$), *Ptk2b* (+31.3-fold, q = 0.0002), and *Plk2* (+8-fold, q = $5.2e^{-6}$) (*Boda et al., 2004*; *Seeburg et al., 2008*; *Bartos et al., 2010*). Therefore, a number of upregulated genes are associated with elevated excitatory synaptic input, potentially contributing to synaptic plasticity and increased AGRP neuron activity previously reported for FD mice (*Takahashi and Cone, 2005*; *Yang et al., 2011*; *Liu et al., 2012*).

During food deprivation, elevated AGRP neuron activity results in increased neurotransmitter and neuropeptide release (*Atasoy et al., 2012*). In line with correspondingly high *Agrp* and *Npy* expression (*Figure 1C* and *Figure 1—figure supplement 1D*), transcripts for neuropeptide secretory

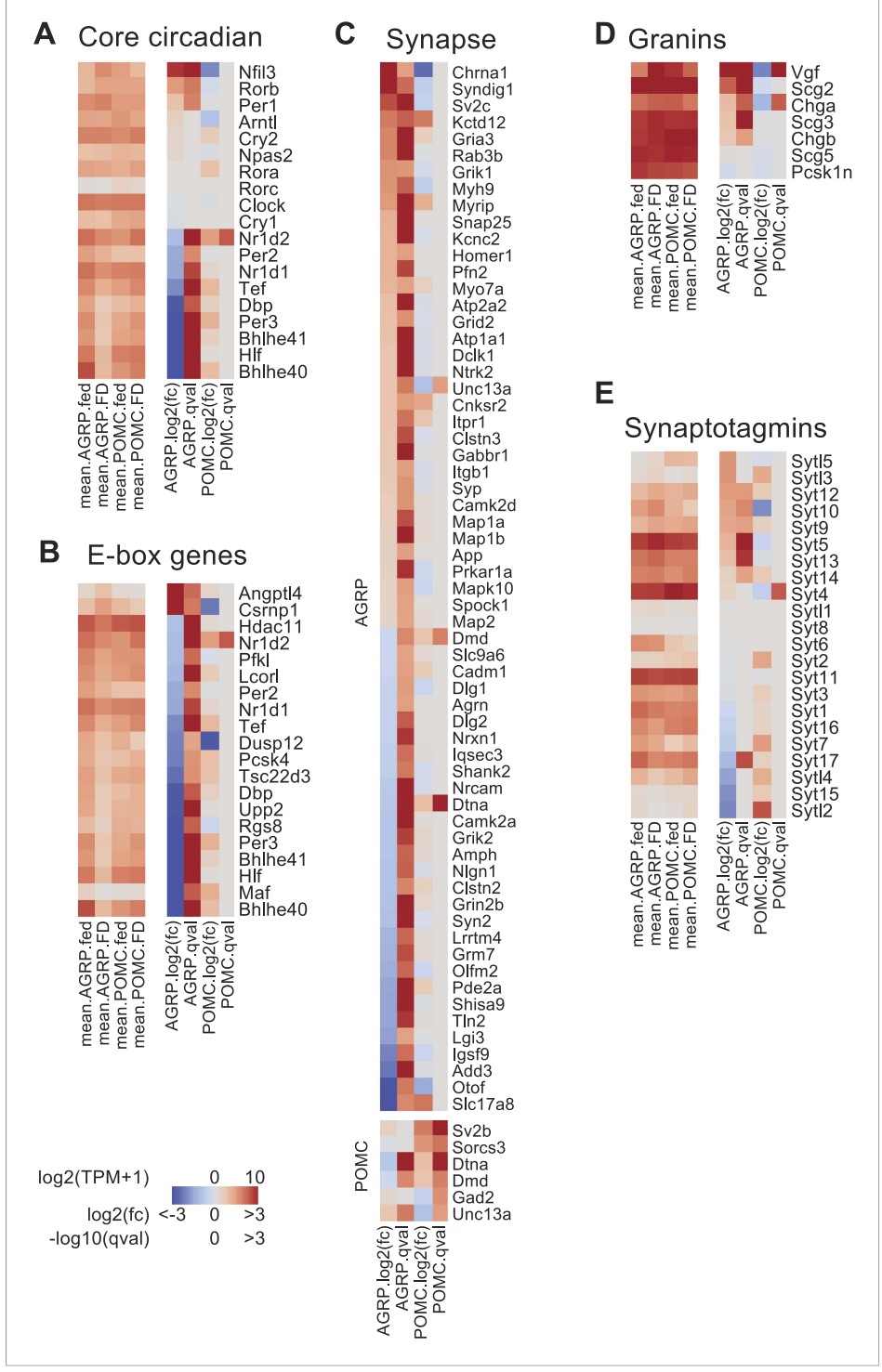

**Figure 3**. Changes in the expression of circadian and synapse-associated genes after food-deprivation. (**A**) Gene expression changes for core circadian genes. Left, mean expression level [log2(TPM)] of each transcript of each experimental group. Right, log2(fold-change) and q-values for differential expression between FD and fed states separately for AGRP and POMC neurons. (**B**) E-box genes differentially expressed during food-deprivation. (**C**) DEG with synapse-localized functions (Gene Ontology: 0045202). Top, AGRP neurons: FD vs fed; bottom, POMC neurons FD vs fed. (**D**, **E**) Gene expression changes for granin genes (**D**) and synaptotagmins (**E**).

The following figure supplement is available for figure 3:

**Figure supplement 1**. Arntl/Bmal expression in AGRP neurons.

vesicle-associated granin-family molecules were selectively upregulated in AGRP neurons (*Figure 3D*). Moreover, a range of synaptotagmin-family transcripts, which mediate different aspects of calcium-dependent vesicle release, are selectively regulated in AGRP neurons by food deprivation (*Figure 3E*). For example, *Syt5*, *Syt9*, and *Syt10* are upregulated in AGRP neurons from FD mice, and these gene products localize to peptidergic vesicles and regulate activity-dependent peptide release (*Cao et al., 2011*). GABA signaling is also important in AGRP neurons, and synaptic vesicle glycoprotein 2C (*Sv2c*, +5.4-fold, q = 5e$^{-10}$), which regulates the readily releasable pool (*Xu and Bajjalieh, 2001*) is increased, as is *Snap25* (+2.1-fold, p = 1.5e$^{-6}$), a key SNARE complex component responsible for vesicle fusion. Collectively, these changes in gene expression show some of the molecular underpinnings for processes that mediate increased AGRP neuron output in energy deficit due to alterations of excitatory synaptic inputs, synaptic plasticity, as well as elevated GABA and neuropeptide release.

## Ion channels

AGRP and POMC neuron electrical properties are determined by expression of distinct groups of ion channels. Despite the importance of neuron electrical activity for influencing appetite and body weight, only a few ion channel subunits have been determined in these cell types and little is known about their regulation by energy deficit state. Previous work has shown transient receptor potential C (TRPC) ion channel subunit expression in POMC neurons (*Qiu et al., 2010*), which is confirmed by our RNA-Seq data, and we find that an overlapping set of TRPC channels (*Trpc1*, *Trpc3-Trpc7*) is also expressed in AGRP neurons (*Figure 4A*). Moreover, *Kcnq3* is downregulated in AGRP neurons with food-deprivation (*Figure 4A*), as previously reported (*Roepke et al., 2011*). Thus, these RNA-Seq data are consistent with established changes in ion channel expression.

This transcriptional profiling resource also identifies other potentially important ion channel genes that cell type-selectively change expression with energetic state and have not been previously investigated in AGRP and POMC neurons (*Figure 4A–C*). One of the most striking changes in ion channel gene expression in AGRP neurons during food-deprivation was a sharp reduction of *Kcnn3* (also called *Sk3*, −5.4-fold, q = 0.0006), which is a small conductance calcium-activated potassium channel (SK) that attenuates action potential firing rate during elevated activity. Other SK channels were not appreciably expressed in AGRP neurons (*Figure 4A*). Electrophysiological characterization of AGRP neurons showed the presence of SK-mediated tail currents, confirmed by blockade with the highly selective SK channel antagonist, apamin (*Figure 4D*). Notably, the apamin-sensitive SK conductance was absent in FD mice (*Figure 4G*), consistent with reduced *Kcnn3* expression. Apamin blockade of SK channels in AGRP neurons resulted in higher firing rates (*Figure 4E,F*), which often elicited bursts and plateau potentials (*Figure 4—figure supplement 1A*). Similarly, AGRP neurons from FD mice showed a comparable increase in excitability, but in this state the neurons were not sensitive to apamin, consistent with greatly reduced SK-channel expression (*Figure 4H,I* and *Figure 4—figure supplement 1B*). Thus, by starting from cell type-specific RNA-Seq analysis, we found that *Kcnn3* regulates firing rate as well as burst firing in AGRP neurons, and its reduction with food-deprivation plays an important role in increasing the excitability of AGRP neurons.

In addition to the examples highlighted here, this resource shows many other ion channels and regulatory subunits that are altered by energy deficit in AGRP neurons, including ionotropic glutamate receptors, GABA receptors, the ionotropic ATP receptor *P2rx4*, sodium channels, calcium channels, additional potassium channels and others (*Figure 4A–C*). This resource provides a list of ion channels expressed in AGRP and POMC neurons, and it is a foundation for a concrete understanding of the electrical properties of AGRP and POMC neurons in basal and energy deprived states.

## GPCRs

GPCRs are critical molecular control points for neuronal function. AGRP and POMC neurons express multiple GPCRs, many of which respond to circulating hormones, neuropeptides, or neurotransmitters. RNA-Seq provides a cell type-specific taxonomy of GPCRs expressed under different conditions. For each cell type, many GPCRs were detected at expression levels greater than 20 transcripts per million (TPM) under either fed or FD conditions (AGRP: 59, POMC: 61, AGRP or POMC: 80) (*Figure 5A*), which is more than previous estimates of GPCRs for hypothalamic regulation of energy homeostasis (*Schioth, 2006*). Also, several GPCRs show high differential expression between AGRP and POMC neurons: 8 GPCRs were >10-fold differentially expressed in AGRP neurons and 13 GPCRs were >10-fold

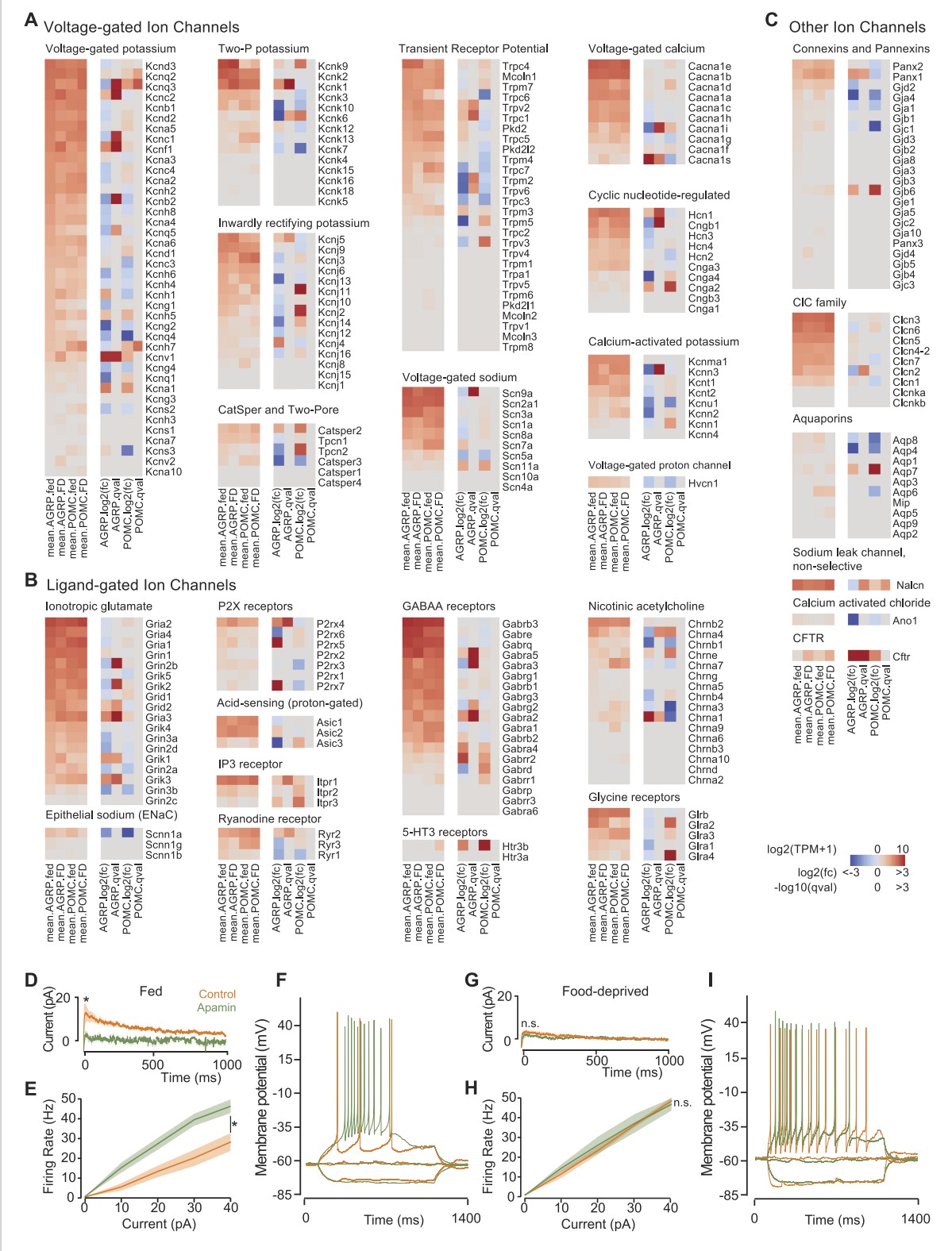

**Figure 4**. Ion channel gene expression in AGRP and POMC neurons. (**A–C**) Gene expression for voltage-gated ion channels (**A**), ligand-gated ion channels (**B**), and other ion channels (**C**). For each colormap, left, mean expression level [log2(TPM)] of each transcript of each experimental group. Right, log2(fold-change) and q-values for differential expression between FD and fed states separately for AGRP and POMC neurons. (**D**) Tail currents elicited in AGRP neurons from fed mice by a voltage step from −30 mV to −60 mV in the absence (n = 5) and presence of apamin (n = 5). Unpaired t-test. *p < 0.05. Lines show mean, shading

*Figure 4. Continued*

shows s.e.m. (**E**) Firing rate from current injection to AGRP neurons from fed mice in the absence (n = 8) and presence (n = 13) of apamin. Unpaired t-test. (**F**) Example of action potential firing in AGRP neurons from fed mice in response to −10, 0, +10 pA in the absence and presence of apamin. (**G**) Tail currents in AGRP neurons from FD mice (−apamin, n = 10; +apamin, n = 5). Unpaired t-test. n.s., p > 0.05. (**H**) Firing rate from AGRP neurons from FD mice (−apamin, n = 4; +apamin, n = 4). Unpaired t-test. (**I**) Example of action potential firing in AGRP neurons from FD mice (current injections: −10, 0, +10 pA).

The following figure supplement is available for figure 4:

**Figure supplement 1**. Burst firing in AGRP neurons with apamin or food deprivation.

overrepresented in POMC neurons (*Figure 5A*). Therefore, differential GPCR expression can separately regulate the function of AGRP and POMC neurons.

GPCR expression was significantly overrepresented in genes that were differentially expressed after food deprivation in AGRP and POMC neurons (AGRP: p = 0.0002, POMC: p = 3.3e$^{-5}$, hypergeometric-test). For AGRP neurons, 13 genes were >twofold upregulated and 11 genes were >twofold downregulated by food deprivation (*Figure 5B*). All significantly upregulated GPCRs were either G$_q$-protein coupled (e.g., *Ghsr*, *Nmur2*, *Gpr83*, *Htr2a*, *Nmbr*, *Hcrtr2*) or G$_s$-protein coupled receptors, several of which show high ligand-independent basal G$_s$-protein activity (*Gpr3*, *Gpr6*, *Gpr64*). The most strongly downregulated genes were G$_i$-protein coupled receptors (*Npy2r*, *Hrh1*, *Hrh3*, *Htr1a*). Conversely, POMC neurons showed the opposite pattern with six differentially regulated GPCR transcripts where all upregulated genes were G$_i$-protein coupled (4/6) and the downregulated genes were G$_q$- or G$_s$-protein coupled (2/6). For example, two GPCRs were reciprocally regulated in AGRP and POMC neurons with food-deprivation, most strikingly G$_i$-protein-coupled *Hrh3* (AGRP: −9.5-fold, q = 4.7e$^{-8}$; POMC: +5.3-fold, q = 0.003; *Figure 5—figure supplement 1*). Taken together, these changes reveal an opposite response pattern during food deprivation for GPCR expression levels in AGRP and POMC neurons, highlighting the importance of cell type-specific transcriptional profiling for assessing gene expression regulation in the brain. Because G$_q$-protein signaling increases AGRP and POMC neuron activity and G$_i$-protein signaling reduces activity in these cell types (*Krashes et al., 2011*; *Atasoy et al., 2012*), the pattern of GPCR expression is consistent with elevated AGRP neuron activity and reduced POMC neuron activity during food deprivation.

In addition to G$_q$- and G$_i$-protein coupled signaling, elevated G$_s$-protein coupled signaling was prominent in AGRP neurons, especially upregulation of *Gpr3*, *Gpr6*, and *Gpr64*, which are constitutively active G$_s$-coupled receptors (*Uhlenbrock et al., 2002*). Moreover, the G$_s$ subunit *Gnas* was upregulated selectively in AGRP neurons (AGRP: +1.8-fold, q = 0.0002, POMC: −1.1-fold, q = 0.73). However, the physiological consequences of G$_s$-protein coupled signaling in AGRP neurons are not well understood. We extended our RNA-Seq observations by examining *Gpr6* expression using quantitative smFISH, which showed a significant increase in the number of *Gpr6* transcripts in AGRP neurons after food deprivation (p = 4.8e$^{-10}$, ks-test, *Figure 5C–E*). To test the functional consequence of elevated *Gpr6* expression in AGRP neurons, we transduced *Agrp*$^{Cre}$ mice with a Cre-dependent virus co-expressing *Gpr6* and a fluorescent protein (AGRP$^{Gpr6}$ mice; *Figure 5F–H*). AGRP$^{Gpr6}$ mice showed significantly elevated body weight compared to mice expressing a fluorescent protein alone (*Figure 5I*). These experiments reveal a potential role for *Gpr6* and G$_s$-coupled signaling in AGRP neurons for positive regulation of body weight.

## Secreted peptides

Neuropeptide expression is typically used to discriminate AGRP and POMC neurons. Transcriptional profiling shows that these two cell populations are distinguished by several additional secreted proteins (*Figure 6A*). Neuropeptides were also among the most highly regulated genes in response to food-deprivation in AGRP neurons (*Figure 6B*). Several neuropeptide transcripts with increased expression in our dataset have been previously shown to increase appetite, for example, *Agrp*, *Npy*, *Vgf*, *Pdyn*. We also observed increased expression of peptides that are associated with reduced appetite, based on pharmacological experiments: *Nmb* (+8.0-fold, q = 6.0e$^{-5}$), *Nts* (+107-fold, q = 0.0001), *Nucb2* (+2.3-fold, q = 0.0002). However, these neuropeptides may have a local signaling role that promotes appetite. For example, *Nmbr* (+8.8-fold, q = 0.0008) and *Ntsr1* (+41-fold, q = 0.005) were

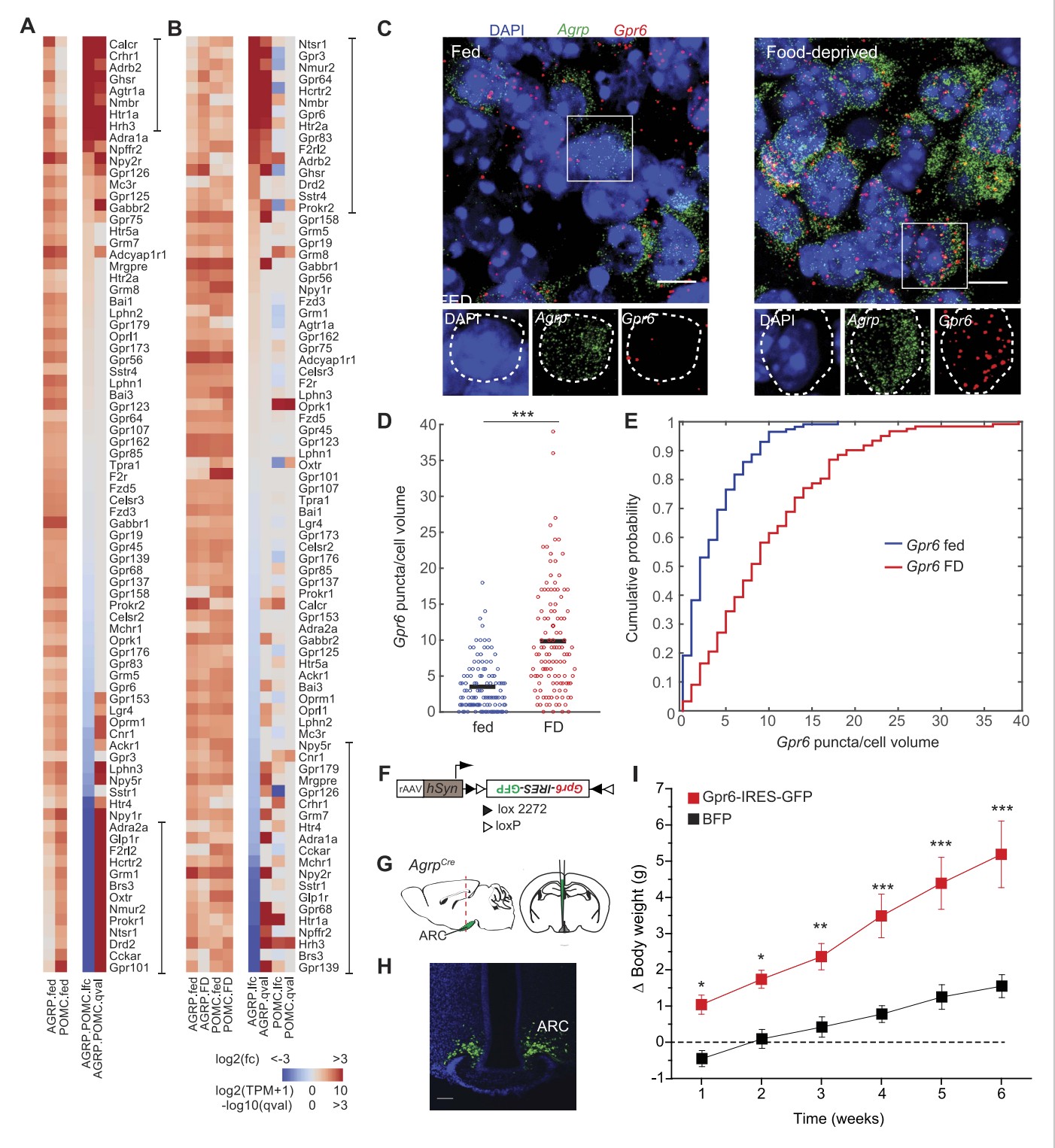

**Figure 5**. G-protein coupled receptors regulated by food-deprivation. (**A**, **B**) All GPCR genes expressed (TPM > 20) in at least one group, sorted by log2 (fold-change) between AGRP and POMC neurons (**A**) or AGRP or POMC neurons FD/fed (**B**). Bars indicate genes with >10-fold change (**A**) or >twofold change (**B**). (**C**) Double smFISH for *Agrp* and *Gpr6*. Scale, 10 μm. (**D**, **E**) Population counts (bars: mean values) (**D**) and cumulative probability distributions (**E**) of *Gpr6* puncta per cell volume in AGRP neurons (p = $4.8e^{-10}$, ks-test). Fed, n = 115; FD, n = 122; 3 mice per condition. (**F**) Cre-dependent viral vector for cell type-specific *Gpr6* overexpression in AGRP neurons. hSyn: synapsin promoter. Black and white triangles denote heterotypic loxP sites for stable inversion of *Gpr6-IRES-GFP*. (**G**) Schematic for viral transduction and cell-type specific overexpression of *Gpr6* in *Agrp^Cre* mice. (**H**) Representative image
*Figure 5. continued on next page*

*Figure 5. Continued*

showing *Gpr6-IRES-GFP*-transduced AGRP neurons. Scale, 100 μm. (**I**) Body weight change from pre-injection weight (starting age: 8 weeks) in *Agrp*[Cre] mice expressing *Gpr6-IRES-GFP* or *BFP* (2-way ANOVA, one factor repeated measures, transgene: $F_{1,65} = 19.6$, $p < 0.001$; time: $F_{5,65} = 30.1$, $p < 0.001$; interaction: $F_{5,65} = 4.2$, $p = 0.002$). Holm-Sidak correction for multiple comparisons. AGRP[Gpr6] n = 9 mice, AGRP[BFP] n = 7 mice. Data is mean ± s.e.m. *p < 0.05, **p < 0.01, ***p < 0.001.

The following figure supplement is available for figure 5:

**Figure supplement 1**. Opposite differential expression of *Hrh3* in AGRP and POMC neurons after food deprivation.

also strongly upregulated in AGRP neurons, are $G_q$-coupled receptors, and *Nmb* has been previously demonstrated to increase AGRP neuron electrical activity (*van den Pol et al., 2009*).

Other genes were upregulated by energy deficit that have not been previously associated with appetite regulation, for example *Ccl17* (+21.6-fold, q = 0.0012) and *Thbs1* (+12.4-fold, q = 0.0002). We further investigated the chemokine CCL17, which was highly upregulated in AGRP neurons. CCL17 is a 103 aa member of the CC chemokine group with no previously reported functional characterization in the central nervous system. Assessment of *Ccl17* expression using double smFISH in AGRP neurons confirmed food deprivation-induced upregulation (p = $1.4e^{-20}$, ks-test, *Figure 6C–E*). Microinjection of recombinant CCL17 (500 ng) into the lateral cerebral ventricle elicited a small increase of food intake during the light period (1-hr: p = 0.02, 2-hr: p > 0.05; rank sum-test, *Figure 6F*). More strikingly, chronic viral overexpression of *Ccl17* and GFP in adult mice by selective expression in AGRP neurons (AGRP[Ccl17] mice) resulted in a progressive elevation of weight gain compared to mice expressing a fluorescent protein alone (*Figure 6J*), providing preliminary evidence that CCL17 may play a role in regulating body weight.

Conversely, we considered the possibility that genes coding for secreted proteins with anorexigenic properties were selectively repressed in AGRP neurons after food-deprivation. For example, *Fgf1*, which has been shown to reduce appetite (*Sasaki et al., 1991*), was strongly downregulated (−4.1-fold, q = 0.0004). To explore this idea, we tested a number of secreted proteins that were downregulated in AGRP neurons after food deprivation and previously had not been examined for regulation of food intake. Based on these criteria, we selected four peptides for further analysis: Pleiotrophin (*Ptn*, −2.3-fold, q = 0.005), a heparin-binding cytokine, which inhibits receptor protein tyrosine phosphatase β/ζ; autotaxin (*Enpp2*, −2.2-fold, q = 0.00013), a secreted enzyme that converts lysophosphatidylcholine into the lipid second messenger lysophosphatidic acid; cerebellin 4 (*Cbln4*, −7.6-fold, q = $2.9e^{-5}$), a transneuronal regulator of synaptic function; and bone morphogenic protein 3 (*Bmp3*, −17.7-fold, q = $2.0e^{-5}$), a member of the TGFβ superfamily (*Figure 6B*). Based on smFISH, we confirmed that *Cbln4 and Bmp3 were* downregulated in AGRP neurons after food-deprivation (*Cbln4*: p = $6.8e^{-17}$, *Bmp3*: p = e −12, ks-test; *Figure 7A–F*). To test whether these proteins influence food intake, we delivered them by intracerebroventricular injection. All four proteins significantly reduced food intake over a 24-hr period (*Figure 7G*), consistent with reduction of their expression levels during food deprivation. Therefore, cell type-specific RNA-Seq is also effective for identifying new secreted proteins that regulate appetite.

## Discussion

The molecular processes governing the function of AGRP and POMC neurons have been a central focus for understanding energy homeostasis and for identifying new approaches to influence appetite and body weight in humans. This cell type-specific RNA-Seq resource from AGRP and POMC neurons reveals an extensive program of gene expression changes selectively in AGRP neurons during energy deficit associated with increased protein translation and folding, circadian gene expression, increased neuronal activity and synaptic release of neurotransmitter and neuropeptides, and alterations in secreted protein and cell surface receptor expression. POMC neurons show a much smaller number of changes, and some of these are associated with reduced activity, as would be expected from suppression of POMC neuron function under energy deficit conditions. Therefore, AGRP neurons are much more sensitive to energy deficit states than POMC neurons.

Differential gene expression in AGRP neurons after food deprivation is also considerably greater than has been observed in other cell types after various perturbations of brain state. *Miller et al. (2011)*

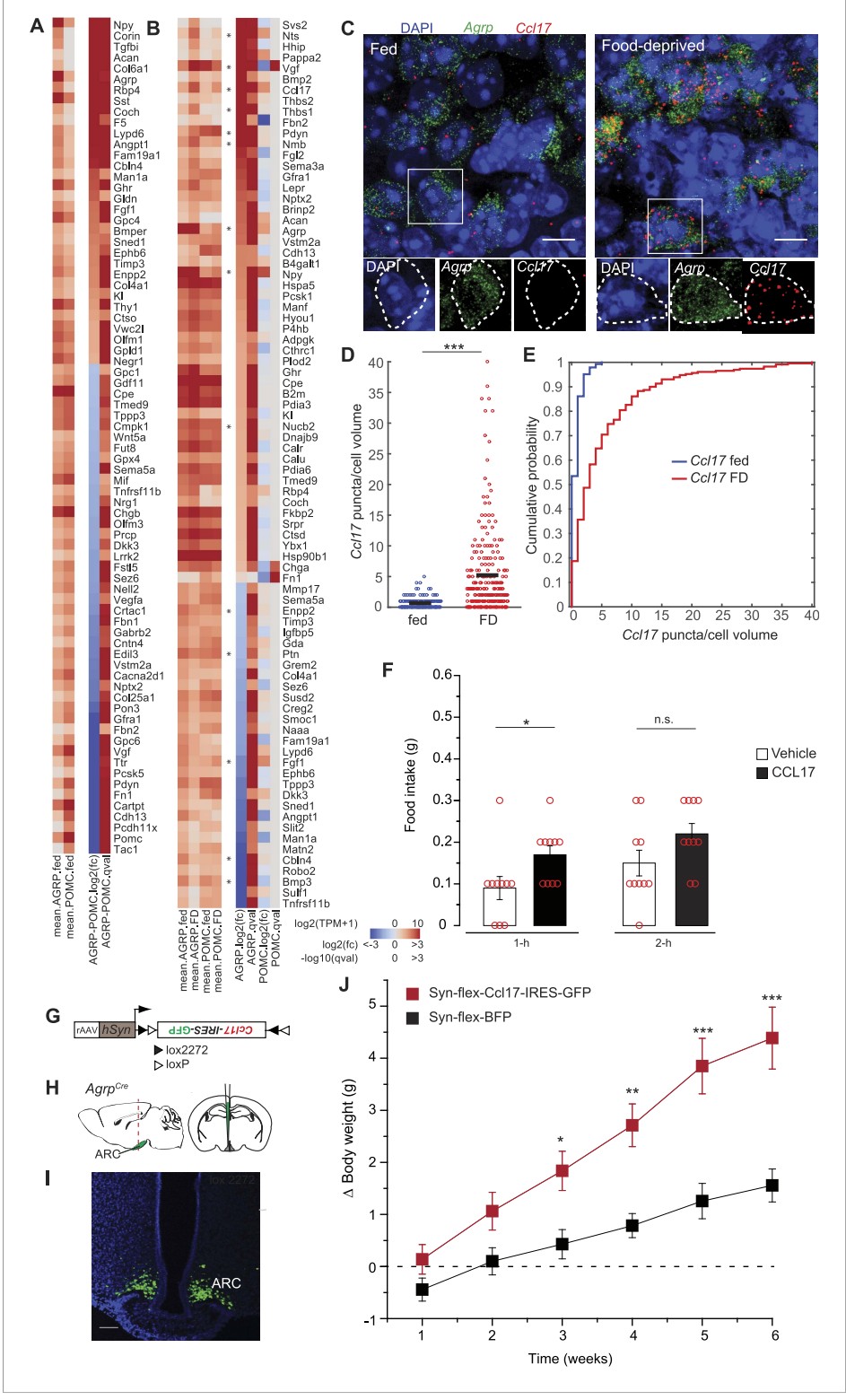

**Figure 6**. Secreted proteins regulated by food-deprivation. (**A**, **B**) Secreted protein genes differentially expressed between AGRP and POMC neurons (**A**) or AGRP or POMC neurons FD/fed (**B**). Genes mentioned in the text are labeled with an asterisk. (**C**) Double smFISH for *Agrp* and *Ccl17*. Scale, 10 μm. (**D**, **E**) Population counts (bars: mean values) (**D**) and cumulative probability distributions (**E**) of *Ccl17* puncta per cell volume in AGRP neurons (p = 1.5e$^{-20}$, ks-test). Fed, n = 144; FD, n = 230; 3 mice per condition. (**F**) Food intake at start of light period 1-hr and 2-hr after

*Figure 6. continued on next page*

*Figure 6. Continued*

intracerebroventricular injection of either saline or recombinant CCL17 (500 ng). Rank-sum test. (**G**) Cre-dependent viral vector for cell type-specific *Ccl17* overexpression in AGRP neurons. *hSyn*: synapsin promoter. Black and white triangles denote heterotypic loxP sites for stable inversion of *Ccl17-IRES-GFP*. (**H**) Schematic for viral transduction and cell-type specific overexpression of *Ccl17* in the brains of *Agrp^Cre* mice. (**I**) Representative image showing *Ccl17-IRES-GFP*-transduced AGRP neurons. Scale, 100 μm. (**J**) Body weight change from pre-injection weight (starting age: 8 weeks) in *Agrp^Cre* mice expressing *Ccl17-IRES-GFP* or *BFP* (2-way ANOVA, one factor repeated measures, transgene: $F_{1,65}$ = 12.0, p = 0.004; time: $F_{5,65}$ = 14.8, p < 0.001; interaction: $F_{5,65}$ = 14.8, p < 0.001). *BFP* data is same as *Figure 5I*. Holm-Sidak correction for multiple comparisons. AGRP^Ccl17 n = 9 mice, AGRP^BFP n = 7 mice. Data is mean ± s.e.m. n.s., p > 0.05, *p < 0.05, **p < 0.01, ***p < 0.001.

compared gene expression in fast-spiking GABAergic interneurons from 48-hr muscimol-treated and saline-treated sides of motor cortex, which led to dramatic alteration in firing properties of these neurons but only 13 DEG (q < 0.05). We observed >160-fold more DEG in AGRP neurons after food-deprivation using the analysis criteria as *Miller et al. (2011)*. Another strong neuronal perturbation, genetic ablation of the important transcriptional repressor *Mecp2*, only resulted in 10–50% of the number of DEG in various cell types (*Sugino et al., 2014*) compared to AGRP neurons from FD mice using the same analysis criteria. Finally, RNA-Seq analysis of motor neurons in an amyotrophic lateral sclerosis mouse model showed only 62 genes differentially expressed (q < 0.05) relative to wildtype motor neurons (*Bandyopadhyay et al., 2013*), which is 34-fold fewer differentially expressed transcripts than from AGRP neurons under food deprivation using the same analysis criteria. The high number of DEG in our dataset is influenced by our study's greater statistical power, use of RNA-Seq (instead of microarrays), as well as high purity and more narrowly defined cell types. However, the relatively small number of DEG measured in POMC under similar experimental conditions indicates that the magnitude of changes in gene expression in AGRP neurons is also based on a selectively tuned response in these neurons to the energy deficit physiological state. Therefore, our results show that neuronal gene expression in vivo can have much greater dynamic changes from alteration of genetic, physiological, or behavioral state than previously reported.

Many DEG were consistent with ER-stress responses associated with the transition to elevated protein translation in AGRP neurons during periods of increased neuropeptide release. Indeed, secreted proteins are among the most abundant transcripts in AGRP neurons and their abundance increases further during energy deficit. We observed increased splicing of the UPR regulator Xbp1s, as well as multiple protein folding chaperones, transcripts encoding protein degradation machinery, and oxidative stress signaling molecules during food deprivation. This adaptive cellular response to food deprivation in AGRP neurons is different to what has been found using whole hypothalamus analysis, where UPR signaling has been primarily associated with overnutrition states (*Ozcan et al., 2009*). Interestingly, overexpression of Xbp1s in POMC neurons has been shown to facilitate the function of those neurons (*Williams et al., 2014*), and a similar effect may be operating in AGRP neurons. In addition, we found that anti-apoptotic pathways were also prominently increased in AGRP neurons during food-deprivation, which is also indicative of considerable cellular stress and the importance of protecting against cell death for this critical energy homeostasis neuron population.

We also found reduced expression of E-box-regulated circadian genes during energy deficit in AGRP neurons. AGRP neurons contribute to increased locomotor activity during scheduled feeding (*Tan et al., 2014*), which is associated with the food-entrained circadian rhythm. Moreover, E-box genes are expressed primarily during the light period (*Rey et al., 2011*), when mice normally eat little, and suppression of these genes is similar to the expression level of these genes at night, a time when mice consume the most food. The cause and downstream consequences of this 'night-like' pattern of circadian gene expression are an important area for further investigation into the circadian control of AGRP neurons, but transcriptional repression involving *Sfpq* (*Psf*) (*Duong et al., 2011*) is a candidate molecule for this pathway.

GPCR gene expression was also substantially altered by food-deprivation in both AGRP and POMC neurons. For AGRP neurons, this was largely associated with $G_q$- and $G_s$-coupled GPCR upregulation as well as reduced $G_i$-coupled GPCRs. This is in line with activation of AGRP neurons by $G_q$-protein-coupled signaling pathways through the $G_q$-coupled DREADD hM3Dq (*Krashes et al., 2011*).

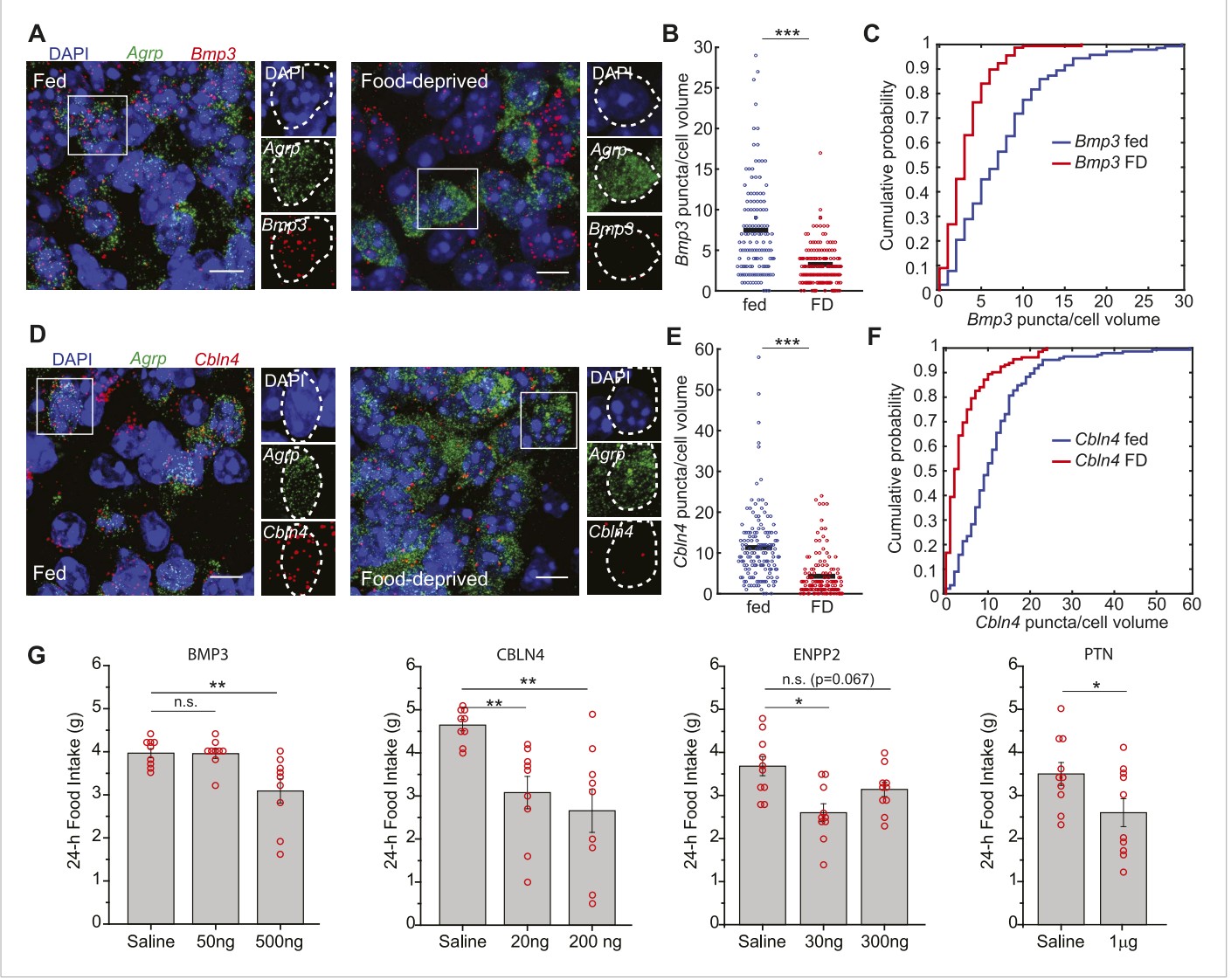

**Figure 7**. Secreted proteins that are downregulated in AGRP neurons with food-deprivation reduce food intake. (**A–F**) Double smFISH for *Agrp* and (**A**) *Bmp3* or (**D**) *Cbln4*. Scale, 10 μm. Population counts (**B**, **E**) and cumulative probability distributions (**C**, **F**) for *Bmp3* and *Cbln4* (p = $9e^{-12}$ and p = $6.8e^{-17}$, ks-test). *Bmp3* fed, n = 142; *Bmp3* FD, n = 157 cells; *Cbln4* fed, n = 145; *Cbln4* FD, n = 132; 3 mice per condition. (**G**) Mean food intake (24 hr) after intracerebroventricular injection of either saline or recombinant BMP3 (ANOVA, $F_{2,24}$ = 7.5, p = 0.003), CBLN4 (ANOVA, $F_{2,24}$ = 8.0, p = 0.002), ENPP2 (ANOVA, $F_{2,27}$ = 7.3, p = 0.003), or Pleiotrophin (unpaired t-test). Holm-Sidak correction for multiple comparisons. Data is mean ± s.e.m. n.s., p > 0.05, *p < 0.05, **p < 0.01, ***p < 0.001.

In addition, overexpression of the constitutively active $G_s$-protein-coupled *Gpr6* in AGRP neurons led to a significant increase in body weight. A similar role may extend to other constitutively active $G_s$-protein-coupled receptors that are upregulated by energy deficit, such as *Gpr3* and *Gpr64*, and this effect is likely enhanced by the concomitant fall in $G_i$-protein coupled receptor expression. Interestingly, we noted that *Hrh1* and *Hrh3* are sharply reduced in AGRP neurons during energy deficit, indicating reduced responsiveness to histamine. Because GPCRs are of considerable interest as targets for drug development, this resource of AGRP and POMC neuron-expressed GPCRs provides multiple possibilities for modulating the function of these key energy homeostasis-regulating neuron populations.

Neuropeptide genes were strongly differentially expressed in AGRP neurons. *Ccl17* has not been functionally investigated in the brain, but it was highly increased by food-deprivation as confirmed by

RNA-Seq and smFISH, and overexpression in AGRP neurons increased body weight, consistent with a role in energy homeostasis. Notably, several neuropeptide genes that were strongly downregulated reduced appetite when injected in the lateral cerebral ventricle. For example, the neuropeptide CBLN4 strongly reduced food intake, and *Cbln4* has been implicated in regulating inhibitory synapse function and is thought to play a neuroprotective role (*Chacon et al., 2015*). In addition, PTN is an inhibitor of receptor tyrosine phosphatase β/ζ, and its reduction might reduce cytokine signaling through tyrosine kinase receptors, which include leptin and insulin receptors.

Another important application of cell type-specific RNA-Seq data is to investigate the electrical properties of neurons. A list of expressed ion channels and how they are altered with food-deprivation provides a molecular framework for electrical activity in these neurons. Ex vivo and in vivo recordings of AGRP neuron activity have shown elevated activity and indicated the importance of burst firing in AGRP neurons during energy deficit (*Betley et al., 2015*). Here, we show that an important contributor to AGRP neuron excitability and burst firing in FD mice is reduced expression of the SK3 channel transcript *Kcnn3*. In fed mice, this calcium-activated potassium channel reduces electrical activity during elevated firing, and we find that SK-channel blockade in AGRP neurons promotes burst firing in brain slices.

Taken together, the data described in this resource provide a framework for extensive analysis of energy homeostasis circuits as well as other cell type-specific circuit nodes in the brain. Transcriptomic profiling was performed only on male mice, and future work should examine gene expression in female mice. However, these data allow insights into the cell biology of energy deficit-responsive neurons and highlight the remarkable specificity of gene expression responses of AGRP and POMC neurons. Moreover, in light of the intensive focus on new therapies for obesity as well as undereating disorders (*Gautron et al., 2015*), this detailed resource of molecular components in key energy homeostasis neurons will be valuable to researchers aiming to devise small molecule and peptide modulators of these circuits. In addition, together with the rapid increase in known genetic variants associated with obesity (*van der Klaauw and Farooqi, 2015*), cell type-specific RNA-Seq data from these and numerous other populations essential for energy homeostasis may strengthen understanding of the cell type basis of obesity found in the human population.

## Materials and methods

All experimental protocols were conducted according to U.S. National Institutes of Health guidelines for animal research and approved by the Institutional Animal Care and Use Committee at Janelia Research Campus (protocol 13-92). Experiments conducted in the UK were licensed (PPL 70/7652) under the UK Animals (Scientific Procedures) Act of 1986 following local ethical approval. All surgery was performed under isoflurance anesthesia to minimize suffering.

### Mice

Mice were housed on a 06:00–18:00 light cycle with water and mouse chow ad libitum (PicoLab Rodent Diet 20, 5053 tablet, TestDiet, St. Louis, MO, United States) unless otherwise noted. The following mouse lines were used: *Agrp^Cre* (Jackson Labs Stock 012899, Agrp^tm1(cre)Lowl/J), *Ai9* (ROSA-loxPStoploxP-tdTomato, Jackson Labs Stock 007909), *Pomc^topazFP* (Jackson Labs Stock 008322), and *Npy^hrGFP* (Jackson Labs Stock 006417). Young adult male mice (>6.5 weeks old) were used for experiments. For tdtomato expression in AGRP neurons, *Agrp^Cre* mice were crossed to *Ai9* mice.

### Cell type sorting

AGRP and POMC neurons were obtained by sorting fluorescent neurons form the ARC of *Npy^hrGFP* mice and *Pomc^topazFP* mice, respectively. Male mice (age 6.5–8 weeks) were used for the experiments. Both control and experimental mice were separated into individual fresh cages a day before experiment around noon. For mice in the 24-hr food deprivation condition, only water was provided. On the next day, between 11:00 and 13:00, mice were sacrificed and labeled neurons were manually sorted as described previously (*Hempel et al., 2007*). Briefly, a horizontal hypothalamic slice (300 µm thick) containing the ARC was obtained using Leica Vibratome VT1200S. After digesting with 1 mg/ml Pronase (P5147, Sigma–Aldrich, St. Louis, MO, United States) for 1 hr at room temperature, the ARC was dissected from the slices and triturated using three Pasteur pipettes with decreasing tip sizes in artificial cerebrospinal fluid (ACSF) with 1% FBS (1 ml). Triturated cells were diluted (25 ml) and

poured into a 100 mm Petri dish. After cells settled (5–10 min), labeled neurons were picked using a glass pipette (30–50 µm tip size) and transferred to a clean 35 mm dish containing ACSF with 1% FBS (2 ml). This manual sorting process was repeated two additional times and the final sorted neurons were transferred to a PCR tube containing extraction buffer XB (47 µl) from PicoPure Kit (KIT0204, Life Technologies, Carlsbad, CA, United States), incubated (42°C, 30 min), and the mixture was stored at −80°C until library preparation. The whole process was complete 3–3.5 hr after sacrifice.

## RNA-Seq

RNA was extracted according to PicoPure manufacturer's instructions. Either 1 µl of $10^{-5}$ dilution of External RNA Controls Consortium (ERCC) spike-in control (Life Technologies, #4456740) or (number of sorted cells/50) × (1 µl of $10^{-5}$ dilution of ERCC) was added to the purified RNA and speed-vacced down to 5 µl and immediately processed with reverse transcription by NuGEN Ovation RNA-Seq System V2 (#7102, NuGEN, San Carlos, CA, United States) which yielded 4–8 µg of amplified DNA. This amplified DNA was fragmented to average size of ~200 bp using Covaris E220. Then, NuGEN Encore NGS Multiplex System I kit (NuGEN, 0314) was used to prepare for sequencing with Illumina HiSeq2500. Libraries were sequenced with either fourfold or eightfold multiplexing in one or two lanes. In total six lanes and four sequencing runs were used for the data collection, and lanes were mixed with samples from fed and FD mice as well as from AGRP and POMC neurons. On average 53 million of 100 bp single end reads were obtained per sample (range: 26–97 million reads). Of these, on average 68% mapped uniquely to University of California, Santa Cruz (UCSC) mm10 genome, 2.2% mapped non-uniquely, 4.6% were unmappable and 25% mapped to abundant sequences such as ribosomal RNA, mitochondrial or phiX sequences. RNA-Seq data is available at Gene Expression Omnibus (GEO) (accession number GSE68177).

## RNA-Seq analysis

Adaptor sequences (AGATCGGAAGAGCACACGTCTGAACTCCAGTCAC for Illumina sequencing and CTTTGTGTTTGA for NuGEN SPIA) were removed from de-multiplexed FASTQ format data from Illumina HiSeq2500 using cutadapt v1.7.1 (http://dx.doi.org/10.14806/ej.17.1.200) with parameters '–overlap = 7 –minimum-length = 30'. Then abundant sequences (ribosomal RNA, mitochondrial, Illumina phiX and low complexity sequences) were detected using bowtie2 (*Langmead and Salzberg, 2012*) v2.1.0 with default parameters. The remaining reads were mapped to mm10 genome using STAR (*Dobin et al., 2013*) v2.4.0i with parameters '–chimSegmentMin 15 –outFilterMismatchNmax 3'. Uniquely mapped reads were then assigned to genes using HTSeq (*Anders et al., 2015*) v0.6.1p1 with parameters '-s no -m intersection-nonempty'. For genome annotation, we used GencodeVM4Basic downloaded from UCSC genome browser, which included 35,266 genes. Fragments per kilobase of transcript per million mapped reads (FPKM) for a gene was calculated by dividing counts of reads assigned to the gene by the sum of the length (in kb) of all the exons belonging to the gene and then normalized by library size (in millions). FPKM was then transformed to TPM by dividing by the sum of all FPKM values and multiplying with 1e6. Expression levels in figures are TPM.

ERCC analysis (similar to *Zeisel et al., 2015*) indicated we had 50% detection rate at 22 copy × kb of ERCC spike-ins in a tube. Because we had on average 102 cells in a tube (minimum 44), this suggests we had at least 50% detection rate of all the transcripts larger than 1 kb, even if there was only 1 transcript/cell. We also estimated that 1 TPM corresponded to 3.2 ± 1.9 copies of transcripts/cell from the ERCC data. This was done using a linear fit between log(ERCC TPM) and log(ERCC copy number), ratio of total reads between ERCC and all the genes and number of cells used.

## DEG

To detect DEG, we used limma-voom packages (*Law et al., 2014*; *Ritchie et al., 2015*). Raw read counts for genes with counts per million (CPM) value larger than 1 in at least three samples (16,513 genes) were used as inputs to limma-voom package. Trimmed mean of M-values (TMM) normalization method (*Robinson and Oshlack, 2010*) was used for normalization and the Benjamini and Hochberg method was used for adjusting for multiple tests. Adjusted p-values (q-values) are reported throughout the paper. The log-ratio (coefficients) outputs from limma package was used as adjusted log2(fold-change) [log2(fc)] and reported instead of simple log2(fold-change) calculated from raw TPM values (differences between coefficients and raw TPM fold-change values are due to weights

assignment to each observation calculated by the voom package). Reported DEG were obtained by requiring q-value <0.05, abs[log2(fc) > 1] and mean CPM > 20 in at least 1 cell type/condition. Post-hoc power analysis using the 'RNASeqPower' package for R (*Hart et al., 2013*) indicates that for our sample size (5–6 mice per group), 80% of genes can be detected that have at least a twofold change in expression.

To evaluate this pipeline of DEG calculation, we randomly permuted assignment of four conditions (Agrp.FD, Agrp.fed, Pomc.FD, Pomc.fed) on 21 samples and recalculated DEG. With 1000 permutations, there were $1.7 \pm 25$ (mean $\pm$ sd) DEG for AGRP.fed vs AGRP.FD and $0.77 \pm 16$ for POMC.fed vs POMC.FD and $0.64 \pm 14$ for AGRP.fed vs POMC.fed, yielding empirical false discovery rate (FDR) of 0.2%, 1.6% and 0.1% respectively. The majority of permutations resulted in zero DEG with only biased permutations yielding any DEG.

## Non-neuronal data

To assess contamination of non-neuronal cell types, we used RNA-Seq data by *Zhang et al. (2014a)*. Raw RNA-Seq data was downloaded from GEO (accession number GSE52564) and processed as for the current data. Four specifically and highly expressed genes from each of the previously described non-neuronal cell types (*Zhang et al., 2014a*) were used to assess potential contamination by these cell types.

## Analysis of oligodendrocyte markers in data from *Ren et al. (2012)*

All samples were processed together with apt-probeset-summarize in Affymetrix Power Tools version 1.17.0 with '–a rma-sketch' option. Using 'MoEx-1_0-st-v1.r2.dt1.mm9.core.mps' provided by Affymetrix as meta-probesets (i.e., gene expression value summaries were obtained).

## MDS

MDS was used to visualize gene expression differences between samples. Pseudo-distance, 1 minus correlation coefficient, was used as an input for MDS. Only genes with CPM > 20 in at least 1 cell type/condition were used for this calculation (8198 genes met this criterion).

## Gene annotation enrichment analysis

The dataset with (adjusted) log-fold-change (lfc) and q-values were uploaded to the Ingenuity Pathway Analysis (IPA) server and genes with q < 0.05 and abs(lfc) > 1 (AGRP: 1276, POMC: 53 genes) were used to evaluate pathway enrichment using IPA core analysis option.

## E-box genes

Core circadian reference genes and E-box target genes shown in *Figure 3* were described previously (*Rey et al., 2011*). E-box targets were selected with criteria: conservation score $\geq 0.9$, cycling score (sum of ZT2-ZT22 score) $\geq 160$ and number of associated E-boxes (either E1 or E1–E2) $\geq 1$. Out of 16,513 genes that had read counts in AGRP neurons, 113 were associated with E-boxes, 1346 genes satisfied qval < 0.05 and abs[log2(fc)] > 1 and of these 20 were associated with E-box. From these numbers, overrepresentation probability of food deprivation affected E-box genes was calculated using the hypergeometric test. Even when we did not threshold E-box genes with conservation score and cycling score, E-box overrepresentation in AGRP food deprivation affected genes (qval < 0.05, abs[log2(fc)] > 1) was significant (p = 0.0006). If genes with smaller fold-changes (q < 0.05, abs[log2(fc)] > 0) were included, statistical significance was further improved ($p = 1.5e^{-6}$).

## GPCRs and ion channels

IUPHAR database (*Pawson et al., 2014*) was used to obtain a list of GPCRs and ion channels. Ion channel classification is based on IUPHAR assignment.

## Secreted proteins

Secreted Protein Database (SPD) (*Chen et al., 2005*) and Gene ontology (GO) annotation were used to create a list of secreted proteins. Only genes with confidence level $\leq 2$ from SPD that also intersected with genes which had GO annotation of 'extracellular region' (GO: 0005576) were used. Genes associated with GO: 0005576 were obtained using the QuickGO website.

## ER protein processing genes

KEGG PATHWAY mmu04141 was used to obtain a list of ER-associated genes.

## Synaptic components and transcription factors

QuickGO was used to obtain genes that encode proteins localized to 'Synapse (GO: 0045202)' and proteins with 'Sequence-specific DNA binding transcription factor activity (GO: 0003700)'.

## Kinases and phosphatases

Previously reported lists of mouse kinases (Caenepeel et al., 2004) and phosphatases (Sacco et al., 2012) were used.

## Recombinant adeno-associated viral (rAAV) vectors

To assess intracellular localization of the UPR-related protein ATF6, the DNA encoding enhanced green fluorescent protein (EGFP):ATF6 (Addgene, Plasmid #32955) was inserted into a rAAV2-hSynapsin-FLEX (FLEX: flip-excision) vector in the inverted orientation to create the Cre-dependent viral expression vector (serotype 1) rAAV2/1-hSyn-FLEX-rev-EGFP:Atf6 (6.4e12 Genomic Copies (GC)/ml). For experiments involving Cre-dependent overexpression of Ccl17 or Gpr6 in AGRP neurons, an Origene synthesized fragment containing either Ccl17 or Gpr6 cDNA from mouse was cloned into a rAAV2-hSynapsin-FLEX vector in an inverted orientation, yielding the Cre-dependent viral expression vectors rAAV2/1-hSyn-FLEX-rev-Ccl17-IRES-EGFP (3.3e12 GC/ml), and rAAV2/1-hSyn-FLEX-rev-Gpr6-IRES-EGFP (7.1e12 GC/ml). Cre-dependent expression of blue fluorescent protein (BFP) used rAAV2/1-hSyn-FLEX-rev-BFP (1.9e13 GC/ml). FLEX, Cre-dependent flip-excision switch (Atasoy et al., 2008). Viral vectors were produced by the Janelia Farm Molecular Biology Core Facility.

## Viral injections and cannula placement

For transgene expression in AGRP neurons, male mice 8 weeks of age or older were anaesthetized with isoflurane, and placed into a stereotaxic apparatus (David Kopf Instruments). After introducing a small incision to expose the skull surface, small holes were drilled in skull for immediate viral injections and/or cannula implantation. rAAV was delivered via a pulled glass pipette with diameter between 20 and 40 μm. For targeted rAAV delivery to the ARC, bilateral injections were made at two depths using the coordinates: bregma −1.3 mm; midline ±0.25 mm; dorsal brain surface −6.0 mm and −5.90 mm in Agrp$^{Cre}$ mice. For experiments involving long term (6 weeks) monitoring of body weight small volume injections (50 nl total injected at each of the two sites) were performed to avoid potential long-term effects on body weight associated with large volume viral injections in this brain region. Mice injected with rAAV2/1-hSyn-FLEX-rev-EGFP:Atf6 were used for experiments 7 days post-infection; transgene expression for significantly longer periods of time resulted in aberrant localization of the EGFP signal to the nucleus under baseline conditions.

For experiments involving acute administration of recombinant peptides into the brain, a craniotomy was drilled over the right lateral ventricle and a cannula was implanted at the coordinates: bregma −0.58 mm, midline +1.25 mm, skull surface −2.0 mm; Grip cement (DENTSPLY) was used to anchor the cannula to the skull. For all animal surgeries, postoperative analgesia was provided. Buprenorphine was administered intraperitoneally (0.1 mg/kg) along with ketoprofen administered subcutaneously (5 mg/kg).

## Antibodies

Rabbit anti-GRP78 (also called BiP; 1:4000, Novus Biologicals, San Diego, CA, United States), monoclonal mouse anti-TDP43 (1:32,000, Abcam, San Francisco, CA, United States), guinea pig anti-BMAL1 (1:15,000, Millipore, Billerica, MA, United States), and sheep anti-GFP (1:3,000, AbD Serotec, Raleigh, NC, United States) were used. Species appropriate, fluorophore-conjugated, minimally cross reactive secondary antibodies were obtained from Jackson Immuno (West Grove, PA, United States) and used at a concentration of 1:500. Specificity of anti-BiP antibody was previously verified via siRNA knockdown (Kitahara et al., 2011; Maddalo et al., 2012). Specificity of anti-TDP43 antibody was previously verified via shRNA knockdown (Lee et al., 2015). Specificity of anti-Bmal1 antibody was verified in Bmal$^{-/-}$ tissue (LeSauter et al., 2012).

## Immunohistochemistry and imaging

Mice were transcardially perfused with 4% paraformaldehyde (PFA) in 0.1 M phosphate buffer fixative (pH 7.4). Brains were postfixed in this solution (3–4 hr) and washed overnight in phosphate buffered saline (PBS) (pH 7.4). Brain sections (50 µm thick) were incubated (24–48 hr, 4°C) with primary antibodies diluted in PBS, supplemented with 1% bovine serum albumin (BSA) and 0.1% Triton X-100. Slices were washed three times and incubated with species appropriate secondary antibodies (2 hr, room temperature) and mounted using VECTASHIELD (Vector Laboratories, Burlingame, CA, United States) hard set mounting medium with DAPI (4′,6-diamidino-2-phenylindole). Images were collected by confocal microscopy (Zeiss 510, Zeiss 710, and Nikon A1R), using identical imaging conditions for each experimental group. Fed and FD groups used same conditions as for RNA-Seq data.

Analysis of BiP antibody staining intensity in somatic regions of AGRP and POMC neurons was performed using the software program FIJI on maximal intensity z-projected (10 µm) image stacks. Immunofluorescence intensity was calculated from the average pixel intensity value contained within a 80 pixel × 80 pixel circular region of interest placed over the somatic region of each cell.

TDP43-associated granule abundance in individual AGRP neurons (*Figure 2F*) was measured on 5 µm projections from confocal images acquired with a 63× objective, using the StarSeach analysis tool (http://rajlab.seas.upenn.edu/StarSearch/launch.html) with a threshold setting of 50.

Quantification of nuclear to cytoplasmic ratio of the GFP:ATF6 signal (*Figure 2H*) was performed using FIJI. Nuclear GFP fluorescence intensity was quantified on a per cell basis from a single optical section (as defined by a binarized mask created from the corresponding DAPI signal), then subtracted from the total somatic integrated intensity for each cell (as defined by a binarized mask created from the corresponding tdtomato signal) to yield the uniquely cytoplasmic intensity signal. Nuclear and cytoplasmic values were normalized by area before being used to calculate the relative ratio of nuclear to cytoplasmic fluorescence intensity ratio for each infected cell. To verify efficacy of reporter, rAAV2/1-hSyn-FLEX-rev-EGFP:Atf6 injected mice were implanted with a cannula over the right lateral ventricle and allowed 1 week of post-surgical recovery time before intracerebroventricular injection with either DMSO or the UPR-inducing agent tunicamycin (40 mg/ml, 1 µl total volume). Mice were perfused 24-hr post injection to assess potential alterations in nucleus:cytoplasm GFP ratio.

Initial attempts to quantify somatic Arntl/Bmal immunofluorescence in z-projected confocal stacks or single optical sections produced considerable variability under baseline conditions that was likely due to differences in fixation quality. To ameliorate this issue, we acutely sliced brain sections of 250 µm thickness (using standard brain slicing procedures [*Atasoy et al., 2012*] for electrophysiological recordings), and these were fixed via submersion in 4% PFA in 0.1 M phosphate buffer fixative (pH 7.4) for 1-hr at room temperature. These slices were then washed and subjected to immunofluorescence staining procedures similar to those described earlier. We restricted analysis to the top 20 µm of the slice. Analysis of Arntl/Bmal immunofluorescence intensity in somatic regions of AGRP neurons (identified from $Npy^{hrGFP}$ mice), was performed using custom MATLAB scripts (*Source code 1*) to automatically detect the optical section of maximal intensity in the EGFP fluorescence channel for a given neuron, then to return the average pixel value of the Arntl/Bmal immunofluorescence channel within a 60 pixel × 60 pixel ROI placed over the somatic region of the each cell.

## Electrophysiology

Acute coronal slices (200 µm) were prepared from the ARC of fed and fasted male $Npy^{hrGFP}$ mice (8 weeks old) and incubated at 37°C for 1 hr before being kept at room temperature prior to experiments, in ACSF containing (in mM): $NaCl_2$ 125, KCl 2.5, $NaHCO_3$ 26, $NaH_2PO_4$ 1.25, glucose 25, $CaCl_2$ 2, $MgCl_2$ 1 (pH 7.3 when bubbled with 95% $O_2$ and 5% $CO_2$). Fluorescent cells were visualized on an upright Slicescope (Scientifica, UK) using a 60× objective, and whole-cell patch clamp recordings ($R_{series} < 30$ MΩ) were performed at 35–37°C using a HEKA 800 Amplifier (HEKA, Germany) and borosilicate glass micropipettes with a 3–6 MΩ resistance (Harvard Apparatus, UK) filled with (in mM): K-Gluconate 130, KCl 10, HEPES 10, EGTA 1, $Na_2ATP$ 2, $Mg_2ATP$ 2, $Na_2GTP$ 0.3. Apamin-sensitive tail currents were recorded in the presence of Kynurenic Acid (2 mM, Sigma), Picrotoxin (50 µM, Sigma), TTX (1 µM, Tocris, Minneapolis, MN, United States), ±Apamin (100 nM, Tocris). Current clamp recordings did not contain TTX. Slices were perfused in blockers at least 10 min prior to obtaining recordings, and all comparisons are between populations of cell recorded in the different conditions. Data was analyzed in Python 2.7 using custom written routines.

## Food intake studies

To assess the effect of intracerebroventricular peptide injection on feeding behavior, adult (8–9 weeks) male mice were implanted with a cannula over the right lateral ventricle, and were allowed to recover for 1 week prior to further manipulation. All animals were singly housed for at least 5 days following surgery. Assessment of food intake over 24-hr was performed in home cages with ad libitum access to standard mouse chow. Saline or peptide-containing solution (1 µl total volume) was delivered via a micromanipulator (Narishige) at a speed at 30 nl/min under isoflurane anesthesia. Injections were performed at 17:00. The following commercially available peptides were used for assessment of food intake alterations during the dark cycle: recombinant mouse PTN (R&D Systems, #6580-PL-050, Minneapolis, MN, United States), recombinant mouse ENPP-2/Autotaxin (R&D Systems, #6187-EN-010), recombinant human BMP-3 (R&D Systems, #113-BP-100/CF), and recombinant human CBLN4 (Abnova, #H00140689-PO1). Cohorts of experimental animals were randomly assigned into saline or peptide groups. Mice were given at least 1 day post injection before performing additional manipulations.

Experiments involving acute alterations in food intake were in the early light period and used a similar procedure for dark cycle experiments. Singly housed, cannulated adult male mice (8–9 weeks of age) were injected intracerebroventricularly with either saline or recombinant mouse (R&D Systems, #529-TR-025/CF) at 09:00 and food intake was monitored in the home cage 1 and 2 hr post-injection. Mice received 1 day between experimental sessions before the subsequent injection.

## Long term body weight monitoring studies

To measure the effects of cell-type specific overexpression of *Ccl17*, *Gpr6* and *Bfp*, adult male mice (8–9 weeks of age at start of experiment) were injected with a Cre-dependent rAAV as described above. Body weight was assessed between 11:00 and 13:00 for 6 weeks.

## Fluorescent in situ hybridization

Two-color smFISH was performed on hypothalamus containing fixed frozen sections from male *Agrp*[Cre] mice (8–9 weeks old), using the proprietary probes and methods of Advanced Cell Diagnostics (Hayward, CA, United States) (ACD Technical notes #320535 for tissue prep, and #320293 for Multiplex labeling, http://www.acdbio.com/technical-support/downloads). Fed and FD groups were same conditions as for RNA-Seq data. Briefly animals were anesthetized and sequentially perfused with RNase free solutions of PBS and 4% PFA in PBS. The brains were removed and post-fixed (24 hr, 4°C) in 4% PFA in PBS, incubated in 30% sucrose (12 hr), and the blocked brain was mounted in cryo-embedding media (OCT, Ted Pella, Redding, CA, United States) on a cryostat for sectioning. Frozen sections (15 µm) were mounted on slides, which were air dried (20 min at −20°C) or stored at −80°C for later use. The OCT was washed off with PBS before pretreatment with ACD proprietary reagents PT2 and PT4. After boiling for 5 min in PT2, sections were rinsed in distilled water then ethanol, air dried, and then incubated with ACD proprietary reagent PT4 (30 min, 40°C) in a HybEZ sealed humidified incubator (ACD). We performed dual probe labeling, using probes for *Bmp3* (Mm-Bmp3-C1, #428461-C1), *Cbln4* (Mm-Cbln4-C1, #428471-C1), *Ptn* (Mm-Ptn, custom order), *Hrh3* (Mm-Hrh3-C1, #428481-C1), *Ccl17* (Mm-Ccl17-C1, #428491-C1), and *Gpr6* (Mm-Gpr6-C2, #318251-C2) in one channel and either *Agrp* (Mm-Agrp-C1, #400711-C1; Mm-Agrp-C2, #400711-C2) or *Pomc* (Mm-Pomc-C1, # 314081-C1; Mm-Pomc-C2, #314081-C2) in the complementary channel. Probes were mixed at a 1:50 ratio of Channel 2 and Channel 1 probes. Wax-outlined tissue sections were immersed in Probe mix and incubated (2 hr, 40°C) in the HybEZ humidified incubator, rinsed in ACD Wash Buffer (2 × 2′) then sequentially incubated in ACD proprietary reagents alternating AMP1-FL and AMP3-FL (30 min) with AMP2-FL and AMP4-FL (15 min) with two washes (2 min) between each step. Brain sections were then labeled with DAPI and coverslips were applied. Slides were stored at 4°C before image acquisition at 63× using a Zeiss 710 confocal on an Axio Examiner Z1 upright microscope. Quantification of mRNA particles was performed on cell volumes obtained by maximal intensity projection of 5 mm of tissue acquired with a 63× objective, using the StarSeach Java applet (http://rajlab.seas.upenn.edu/StarSearch/launch.html) with a threshold setting of 50.

## Statistics

For statistical analyses of RNA-Seq data see above. For other experimental data, comparisons were calculated by unpaired or paired two-tail Student's *t*-test, rank sum test, or analysis of variance

(ANOVA). Post hoc multiple comparisons used Holm-Sidak correction. Statistical analyses were performed using Origin, Matlab, or SigmaPlot. Values are means ± s.e.m, unless otherwise noted. n.s., $p > 0.05$, *$p < 0.05$, **$p < 0.01$, ***$p < 0.001$.

## Accession number
RNA-Seq data is available at GEO, accession number GSE68177.

## Acknowledgements

This research was funded by the Howard Hughes Medical Institute (SMS), the NeuroSeq Project Team (KS), Wellcome Trust/Royal Society Henry Dale Fellowship and MRC Grant MC-UP-1201/1 (TB). We thank M Ramirez and K Ritola for molecular biology; M Copeland for assistance with smFISH; K Ritola for virus production; A Zeladonis and S Lindo for mouse breeding and genotyping; L Wang, A Lemire and P Serge for technical help in sorting, library preparation and sequencing; J Takahashi for suggesting consideration of E-box genes; S Eddy and D Stern for comments on the manuscript.

## Additional information

### Funding

| Funder | Grant reference | Author |
| --- | --- | --- |
| Howard Hughes Medical Institute (HHMI) | | Scott M Sternson |
| Wellcome Trust | 098400/Z/12/Z | Tiago Branco |
| Medical Research Council (MRC) | MC-UP-1201/1 | Tiago Branco |

The funders had no role in study design, data collection and interpretation, or the decision to submit the work for publication.

### Author contributions
FEH, KS, Conception and design, Acquisition of data, Analysis and interpretation of data, Drafting or revising the article; AT, Acquisition of data, Analysis and interpretation of data; TB, Acquisition of data, Analysis and interpretation of data, Drafting or revising the article; SMS, Conception and design, Analysis and interpretation of data, Drafting or revising the article

### Ethics
Animal experimentation: All experimental protocols were conducted according to U.S. National Institutes of Health guidelines for animal research and approved by the Institutional Animal Care and Use Committee at Janelia Research Campus under protocol number 13-92. Experiments conducted in the UK were licensed (PPL 70/7652) under the UK Animals (Scientific Procedures) Act of 1986 following local ethical approval. All surgery was performed under isoflurance anesthesia to minimize suffering.

## Additional files

### Supplementary files
• Supplementary file 1. Gene expression data from RNA-Seq. Spreadsheet showing, for each gene, analyzed gene expression data for AGRP and POMC neurons. TPM values for each gene across all samples are provided. Comparisons for AGRP vs POMC gene expression in the fed state(.AgPo) as well as AGRP food-deprived (FD) vs fed (.agrp) and POMC food-deprived (FD) vs fed (.pomc) are also shown. CPM: counts per million, TPM: transcripts per million, fc: fold-change calculated as described in 'Materials and methods', lfc: log2(fc), qval: Benjamini-Hochberg corrected p-value as described in 'Materials and methods'.

• Source code 1. Matlab script for measuring fluorescence intensity from confocal stack image volumes.

## Major datasets

The following dataset was generated:

| Author(s) | Year | Dataset title | Dataset ID and/or URL | Database, license, and accessibility information |
|---|---|---|---|---|
| Sugino K, Henry T, Sternson S | 2015 | Cell type transcriptomics of hypothalamic energy-sensing neuron responses to fasting | http://www.ncbi.nlm.nih.gov/geo/query/acc.cgi?acc=GSE68177 | Publicly available at the NCBI Gene Expression Omnibus (Accession no: GSE68177). |

Standard used to collect data: ARRIVE guidelines.

The following previously published datasets were used:

| Author(s) | Year | Dataset title | Dataset ID and/or URL | Database, license, and accessibility information |
|---|---|---|---|---|
| Accili D, Seigneur EM, Panicker LM, Ren H, Chen C, Simonds WF | 2013 | FoxO1 target Gpr17 activates AgRP neurons to regulate food intake | http://www.ncbi.nlm.nih.gov/geo/query/acc.cgi?acc=GSE45858 | Publicly available at the NCBI Gene Expression Omnibus (Accession no: GSE45858). |
| Zhang Y, Chen K, Sloan SA, Scholze AR, Caneda C, Ruderisch N, Deng S, Daneman R, Barres BA, Wu JQ | 2014 | An RNA-Seq transcriptome and splicing database of neurons, glia, and vascular cells of the cerebral cortex | http://www.ncbi.nlm.nih.gov/geo/query/acc.cgi?acc=GSE52564 | Publicly available at the NCBI Gene Expression Omnibus (Accession no: GSE52564). |
| Cahoy JD, Emery B, Xing Y, Kaushal A | 2007 | A Transcriptome Database for Astrocytes, Neurons, and Oligodendrocytes | http://www.ncbi.nlm.nih.gov/geo/query/acc.cgi?acc=GSE9566 | Publicly available at the NCBI Gene Expression Omnibus (Accession no: GSE9566). |

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
