## [Decision Letter]

Thank you for submitting your work entitled “Cell type-specific transcriptomics of hypothalamic energy-sensing neuron responses to weight-loss” for peer review at *eLife*. Your submission has been favorably evaluated by a Senior Editor and three reviewers, one of whom is a member of our Board of Reviewing Editors.

The reviewers have discussed the reviews with one another and the Reviewing Editor has drafted this decision to help you prepare a revised submission.

Henry and colleagues describe results from a series of transcriptomic profilings of hypothalamic AGRP and POMC neurons. Briefly, the authors manually sorted tagged AGRP and POMC neurons and subjected the cells to RNAseq analysis. Notably, they did this in fed and fasted mice. They provide a comprehensive assessment of gene expression changes, especially in AGRP neurons. They also provide some preliminary functional studies to support the functional significance of some of the changes observed in AGRP neurons. Overall, the studies are well performed and provide abundant novel information. Clearly, the results will be of wide interest. Although several other attempts have been made at collecting and dissecting the most useful transcriptomic information from hypothalamic circuits involved in body weight regulation, this study is offering additional value, mostly due to the scientific rigor applied throughout conceptualization and while conducting these experiments. Several issues remain to be addressed that would strengthen the manuscript:

Overall, the manuscript represents a useful analysis of gene expression, and one which will serve as a useful resource for researchers that study these neurons. Rather than having one or two large/main conclusions, however, the manuscript consists of many “smaller” conclusions that are touched upon throughout the Results and Discussion sections.

A major limitation of the manuscript is the contrast between the comprehensive nature of the gene expression changes vs the somewhat preliminary follow up studies. For example, the authors performed some feeding studies following ICV injections of a few candidate peptides that were altered in AGRP neurons. In addition, the authors overexpressed a few candidates (*Ccl17*, *Gpr6* and *Bfp*) in AGRP neurons using an AAV approach and assessed body weight changes. While it is understandable to provide some in vivo physiological validation/extension of the results, the findings are collectively not nearly as persuasive as the RNAseq data itself. In large part these studies distract from the comprehensive data set provided.

The heat maps as presented are not particularly user-friendly or useful for most readers – nor is their labeling. Converting to a more readable and less formidable format would greatly increase the accessibility of the data.

Another general criticism is the focus of this work on the “AGRP-neurons”, or more precisely the suggestion that those would somehow be the only important neurons governing systemic energy homeostasis. Although undoubtedly these neurons are involved in that process as shown by ablation studies, they are still representing a random group of cells based on their expression of the AGRP protein. Therefore using AGRP as the one and only guiding signal seems like an overestimation. At a minimum, the authors need to briefly discuss this.

---

## [Author Response]

*[…] Overall, the manuscript represents a useful analysis of gene expression, and one which will serve as a useful resource for researchers that study these neurons. Rather than having one or two large/main conclusions, however, the manuscript consists of many “smaller” conclusions that are touched upon throughout the Results and Discussion sections*.

*A major limitation of the manuscript is the contrast between the comprehensive nature of the gene expression changes vs. the somewhat preliminary follow up studies. For example, the authors performed some feeding studies following ICV injections of a few candidate peptides that were altered in AGRP neurons. In addition, the authors overexpressed a few candidates (*Ccl17*,* Gpr6 *and* Bfp*) in AGRP neurons using an AAV approach and assessed body weight changes. While it is understandable to provide some* in vivo *physiological validation/extension of the results, the findings are collectively not nearly as persuasive as the RNAseq data itself. In large part these studies distract from the comprehensive data set provided*.

We agree with the reviewers that there inevitably has to be a large gap between the comprehensiveness of the transcriptomic data and the small number of follow-up experiments that we report. However, the reviewers also acknowledge that it is “understandable to provide some in vivo physiological validation/extension of the results”. This latter point was exactly our intention. We wanted some indication of how predictive some of these findings were, which is why we performed a small number of follow-up experiments. Our interpretation of this critique is that the follow up experiments were called “distracting” because of their preliminary nature, which was not always stated clearly in the text by us. Therefore, we have responded to these reviewer concerns by better qualifying the significance of the follow up experiments and emphasizing their preliminary nature.

In the subsection “Endoplasmic reticulum stress pathways” we state:

“Consequently we examined multiple aspects of ER-stress signaling by analysis of our RNA-Seq data, coupled with preliminary evaluation of ER-stress signaling using immunohistochemistry, a cell type-specific UPR pathway reporter, and smFISH”.

“Taken together, Xbp1-splicing and the patterns of downstream gene expression indicate a previously unreported role for Ern1/Ire1→Xbp1s signaling in the adaptive response of AGRP neurons to energy deficit.”

“Elevated stress granule formation was not detected in AGRP neurons from food-deprived mice (p = 0.76, ks-test, Figure 2), which provides preliminary evidence that Eif2ak3/PERK-mediated translational arrest may not be engaged at this 24-hr food deprivation time-point. Moreover, transcripts for ER protein translocation (*Srp68*, *Srp72*, *Sec61a1, Sec61b1, Sec63, Serp1/Ramp4*) and Golgi trafficking (*Sec14l1*, *Sec22b*, *Sec24d*) were upregulated in AGRP neurons, possibly indicating increased protein translation and folding capacity during energy deficit and consistent with the requirement for increased peptidergic neurotransmission for AGRP neuron function.”

“Together, this group of cell type-selectively upregulated genes is consistent with an adaptive response to increased oxidative stress in AGRP neurons during energy deficit.”

“This pattern of gene expression provides preliminary evidence of a potential role for anti-apoptotic pathways in AGRP neurons during activation in response to energy deficit.”

“Increased neuron activity and elevated neuropeptide production associated with food-deprivation is expected increase the translational-load in AGRP but not POMC neurons, and UPR may serve to cope with elevated neuropeptide and synaptic output.”

In the subsection “Regulation of circadian genes by food-deprivation” we clarify:

“However, detailed examination of pathways that regulate circadian E-box containing genes in AGRP neurons during energy deficit states is required.”

In the subsection entitled “G-protein coupled receptors”, we added the following:

“These experiments reveal a potential role for *Gpr6* and G_s_-coupled signaling in AGRP neurons for positive regulation of body weight.”

Finally, in “Secreted peptides” we stress:

“More strikingly, chronic viral overexpression of *Ccl17* and GFP in adult mice by selective expression in AGRP neurons (AGRP^Ccl17^ mice) resulted in a progressive elevation of weight gain compared to mice expressing a fluorescent protein alone (Figure 6), providing preliminary evidence that CCL17 may play a role in regulating body weight.”

*The heat maps as presented are not particularly user-friendly or useful for most readers – nor is their labeling. Converting to a more readable and less formidable format would greatly increase the accessibility of the data*.

We have taken the approach of using colormaps to highlight aspects of the transcriptomic data that we believe will be of interest to the widest range of researchers in the area of energy homeostasis (ion channels, GPCRs, neuropeptides, etc). Of course, all readers have access to the complete data, but we wanted to provide a guide to key classes of transcripts explicitly in the figures to aid what we expect to be the majority of readers who are in the energy homeostasis field and interested in perusing the data but are not comfortable with downloading the data for deep analysis. Moreover, we know that many people would want to see some of the lists that we proved: e.g. top 30 DEG, all POMC neuron differentially expressed genes (DEG), all GPCR DEG, etc. The colormaps that we provided give easy access to important information that we expect to be used by scientists in the field. However, we are concerned by the reviewers’ critique that we have failed to organize or label these colormaps in an intuitive manner. We believe that the three most essential summary data for each category of gene expression are 1) mean expression level, 2) fold-change of expression level, 3) the q-value for differentially expressed transcripts. This is primarily what is shown in the colormaps, which are the most succinct way to represent these three types of summary statistics in a figure. We feel strongly that this information needs to be easily available to researchers. Therefore, based on the reviewers’ comments, we have worked to improve colormap and colorbar labeling to make reading these maps more intuitive:

a) Only in Figure 1 do we show data in the colormaps for individual samples so that the reader has a feel for the within-group variability. Because expression levels vary for different transcripts, these values are normalized to the max expression for that gene. To reduce confusion, for plots showing normalized expression, this is now noted directly on the figure with “Norm. expression” written at the top of these portions of the figure. Also the “max” column for summary data is now labeled max(TPM), which better indicates that it denotes an expression level (Figure 1 and Figure 1). This issue of showing normalized expression is limited to this figure, and it is necessary to show maximum expression as a reference point for the normalized gene expression. We have also more extensively described the organization of the colormaps in the figure legend, in order to aid the reader.

b) Also, the colorbar labels, which are organized so that one set of colors is needed for all three scales (expression level, fold-change, and q-value), have been reorganized to be more intuitive. Now, the order of the labels reflects the ordering of the data (expression levels, fold-change, q-value). In addition, the label lfc has been changed to log2(fc) so that it exactly corresponds to the colorbar label.

c) For Figure 1, the colorbar label is moved directly over the relevant color maps in order to help the reader know what each scale corresponds to.

d) For Figure 1—figure supplement 2 and all subsequent colormaps in the paper, the left columns show mean expression level of each group and this is now made more clear by including the word ‘mean’ prior to each group label. ‘lfc’ has been replaced with log2(fc), which is now consistent with the label on the colorbar. As above, the colorbar labeling scheme was slightly reorganized so that expression level is shown first, followed by fold change, followed by q value, which now follows to order of the data display in each panels’ colormaps. In addition, the figure legend is now much more explicit in the explanation of the colormap layout.

*Another general criticism is the focus of this work on the “AGRP-neurons”, or more precisely the suggestion that those would somehow be the only important neurons governing systemic energy homeostasis*.

We entirely agree that AGRP neurons are not the only important neurons governing systemic energy homeostasis, and we have taken steps to avoid this unintended impression by altering the first sentence of the Introduction to read: “Neurons that express *Agrp* and *Proopiomelanocortin* (*Pomc*) comprise two intermingled molecularly defined populations in the hypothalamic arcuate nucleus (ARC) that mediate whole-body energy homeostasis *in conjunction with other cell types*.” The final sentence of the paper is also changed to read: “In addition, together with the rapid increase in known genetic variants associated with obesity (78), cell type-specific RNA-Seq data *from these and numerous other populations essential for energy homeostasis* may strengthen understanding of the cell type basis of obesity found in the human population.” However, we are not sure if the reviewers may be concerned about our discussion points highlighting the dramatically greater sensitivity of differential gene expression in AGRP neurons vs POMC neurons. This is not a statement of the relative importance of the two cell types but instead their relative sensitivity to that particular physiological stimulus. We go on to point out how remarkable this is compared to the magnitude of changes found in other examples of cell type-specific RNA-seq after profound neuronal perturbations.

*Although undoubtedly these neurons are involved in that process as shown by ablation studies, they are still representing a random group of cells based on their expression of the AGRP protein. Therefore using AGRP as the one and only guiding signal seems like an overestimation. At a minimum, the authors need to briefly discuss this*.

Our data analysis suggests that the neurons marked by *Agrp* expression are not a random group of cells to the extent that they are clearly distinguished molecularly and functionally from intermingled POMC neurons. This point is made in the cluster analysis of Figure 1—figure supplement 1 and the MDS analysis in Figure 1 and Figure 1—figure supplement 1. Importantly, elimination of *Npy, Agrp*, or *Pomc* from the data still readily allows discrimination of the AGRP and POMC neuron data sets. We go on to show that this even holds if we eliminate all 694 differentially expressed genes between AGRP and POMC neurons (based on the criteria for DEG laid out in the Methods). This means that even the statistically insignificant differences in individual gene expression are collectively still sufficient to distinguish AGRP and POMC neurons. Note that in the third paragraph of the Results section we have written: “Therefore, transcriptional profiles indicate that these canonical markers [referring to *Npy, Agrp*, *Pomc*] are not required to determine cellular identity of AGRP and POMC neurons.” This emphasizes the point that AGRP neurons are not a random cell group defined solely by AGRP expression.